# Dynamic Subspace Estimation from Undersampled Data using Grassmannian Geodesics

## Abstract

This work considers recovering a sequence of low-rank matrices from undersampled measurements, where the underlying subspace varies across samples over time. Existing works involve concatenating all of the samples from each time point to recover the underlying matrix under the assumption that the data are well-approximated by a single, static subspace. However, this assumption is inappropriate for applications where the best low-rank approximations vary over time. To address this issue, we propose a Riemannian block majorize-minimization algorithm that constrains the time-varying subspaces as a geodesic along the Grassmann manifold. Our proposed method can more accurately estimate the best-fit subspaces at each time point, even when there are fewer samples at each time point than the subspace dimension. Theoretically, we show that our algorithm enjoys a monotonically non-increasing objective function while converging to an $\epsilon$-stationary point within $\widetilde{\mathcal{O}}(\epsilon^{-2})$ iterations. We demonstrate the effectiveness of our algorithm on synthetic, dynamic fMRI, and video data, where the samples at each time are either compressed or partially missing.

## 1 Introduction

Subspace estimation and low-rank matrix approximation are fundamental problems that arise in many statistical signal processing and machine learning applications. By modeling the data using a linear subspace, practitioners can efficiently leverage the intrinsic, low-dimensional structure of the data for tasks such as classification (Basri et al., 2007; Hubert & Engelen, 2004), anomaly detection (Huang et al., 2006), and denoising (Zhang et al., 2010), among others. One of the most widely studied and applied methods for subspace estimation is principal component analysis (PCA) (Abdi & Williams, 2010).

In the literature, many existing algorithms, such as PCA, primarily focus on a setting where we wish to fit data with a single linear subspace (Cai et al., 2021; Balzano et al., 2010; 2018b; Mansour & Jiang, 2015; Jain et al., 2013; Chi et al., 2012; Tong et al., 2021; Zhang et al., 2021). In applications where data are collected over time, one may then concatenate data and identify a single static subspace or estimate different low-dimensional approximate subspaces at different time points. However, in applications such as array signal processing (Vaccaro, 2019; Lake & Keenan, 1998a), video processing (Vaswani et al., 2018), graph connectivity analysis Hume & Balzano (2024), and dynamic MRI (Otazo et al., 2015; Babu et al., 2023), the data generation process is dynamic in nature, so assuming the best-approximating subspace is static may be sub-optimal. Despite the abundance of such applications, the literature on dynamic subspace estimation often relies on assumptions such as slow changes or otherwise a static subspace (Narayanamurthy & Vaswani, 2018a; 2019). Furthermore, many works on dynamic subspace estimation assume that the data are fully observed (Saad-Falcon et al., 2024b), limiting applicability for data with missing entries.

In this work, we address these issues by proposing a Riemannian block majorize–minimize (RBMM) algorithm that constrains the time-varying subspaces to evolve along a geodesic on the Grassmann manifold, as illustrated in Figure 1. Instead of assuming a static subspace for low-rank approximation, this geodesic formulation estimates a sequence of subspaces that vary smoothly over time, enforcing smooth transitions between consecutive subspaces. By employing such a model, our algorithm can recover undersampled (miss-

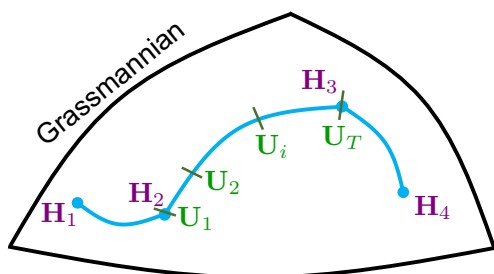

Figure 1: Cartoon illustration of the geodesic model. $\mathbf{H}_1, \ldots, \mathbf{H}_4$ are orthonormal bases that represent subspaces (i.e., points on the Grassmannian), and $\mathbf{U}_i$ are connecting points along a geodesic between two points.

ing or compressed) data more accurately under measurement noise compared to methods that assume a static subspace. We empirically demonstrate the effectiveness of our algorithm in both synthetic and real data settings. For synthetic data, we test our algorithm and comparators on recovering planted geodesic curves with added noise and partial observations. For real data, we test across a wide range of real video and dynamic fMRI reconstruction tasks. Theoretically, we show that our algorithm converges to an $\epsilon$-stationary point in approximately $\widetilde{\mathcal{O}}(\epsilon^{-2})$ iterations, ignoring logarithmic factors, and that the objective function is monotonically non-increasing.

**Notation.** We denote scalars with unbolded letters (e.g., $t, T$), vectors with bold lower-case letters (e.g., $\mathbf{x}$) and matrices with bold upper-case letters (e.g., $\mathbf{X}$). For a matrix $\mathbf{X} = \begin{bmatrix} \mathbf{x}_1 & \cdots & \mathbf{x}_d \end{bmatrix}$ we define $\text{range}(\mathbf{X}) :=$ $\text{span}(\{\mathbf{x}_1, \ldots, \mathbf{x}_d\})$, which is the set of all possible linear combinations of the columns of $\mathbf{X}$. We use $\mathbf{I}_r$ to denote an identity matrix of size $r \in \mathbb{N}$ and $\mathcal{V}^{n \times r} = \{\mathbf{U}^{n \times r} \mid \mathbf{U}^\top \mathbf{U} = \mathbf{I}_r\}$ to denote the set of all $n \times r$ matrices with orthonormal columns. We use $\{\mathbf{U}_i\}$ as shorthand for the set ranging over the index $i$ (i.e., $\{\mathbf{U}_i\}_{i=1}^T$), and similarly for sets over multiple indices (e.g., $\{\mathbf{A}_{i,j}\}$). We use $[L]$ to denote the set $\{1, 2, \ldots, L\}$. Lastly, $\widetilde{\mathcal{O}}(\cdot)$ denotes "big-O" notation that ignores logarithmic factors.

## 2 Problem Formulation

### 2.1 Data Model

Our problem is motivated by a generative model formulation in which the data are generated from a sequence of time-varying, low-dimensional subspaces. Specifically, suppose that at each time point $i \in [T]$, a total of $\ell$ vectors, arranged as columns in a matrix $\mathbf{X}_i \in \mathbb{R}^{n \times \ell}$, are generated from a subspace $\mathbf{U}_i \in \mathcal{V}^{n \times r}$ where $r \leq n$, to form a latent matrix

$$\mathbf{X} = [\mathbf{X}_1 \ldots \mathbf{X}_T] = [\mathbf{U}_1 \mathbf{G}_1 \ldots \mathbf{U}_T \mathbf{G}_T] \in \mathbb{R}^{n \times \ell T}, \tag{1}$$

where each $\mathbf{G}_i \in \mathbb{R}^{r \times \ell}$ is a weighting matrix. Equivalently, we can express this model as

$$\mathbf{X}_i = [\mathbf{x}_{i,1} \ldots \mathbf{x}_{i,\ell}] = [\mathbf{U}_i \mathbf{g}_{i,1} \ldots \mathbf{U}_i \mathbf{g}_{i,\ell}], \qquad \forall i \in [T],$$

where $\mathbf{x}_{i,j} \in \mathbb{R}^n$ and $\mathbf{g}_{i,j} \in \mathbb{R}^r$ denotes the $j$-th column of $\mathbf{X}_i$ and $\mathbf{G}_i$, respectively. Instead of observing $\mathbf{X}_i$ directly, this work considers a challenging setting in which we observe only partial measurements:

$$\mathbf{Y}_i = [\mathbf{y}_{i,1} \ldots \mathbf{y}_{i,\ell}], \quad \text{where } \mathbf{y}_{i,j} = \mathbf{A}_{i,j} \mathbf{x}_{i,j} + \boldsymbol{\eta}_{i,j},$$

where each $\mathbf{A}_{i,j} \in \mathbb{R}^{m \times n}$ is a sensing matrix and $\boldsymbol{\eta}_{i,j} \in \mathbb{R}^m$ is additive noise, typically with $m \ll n$. This setting arises in many signal processing applications; if $\mathbf{A}_{i,j}$ has iid Gaussian entries, we have a compressed sensing setting; if $\mathbf{A}_{i,j}$ are subsampled diagonal matrices, we have the missing data (or matrix completion) setting. Our goal in either of these cases is to estimate all latent vectors $\{\mathbf{x}_{i,j}\}$ given the data $\{\mathbf{y}_{i,j}\}$ and the sensing matrices $\{\mathbf{A}_{i,j}\}$.

However, when $m < n$, the sensing matrix has a non-trivial nullspace, and the problem is under-determined. In the literature, a common approach is to assume a prior (or structure) on each signal $\mathbf{x}_{i,j}$ and use an appropriate regularizer. In this work, since each latent matrix $\mathbf{X}_i$ is assumed to be low-rank, we consider priors that take into account the subspaces $\mathbf{U}_i$ that generate the data. Before presenting our dynamic approach, we first review commonly used static approaches to highlight the differences from our proposed method.

## 2.2 Existing Methods

A common approach to this challenging estimation problem is to drop dependence of the subspace on time, essentially assuming the subspace changes so slowly that it can be ignored. Mathematically, by setting $\mathbf{X} = \mathbf{UG}$ for some subspace $\mathbf{U} \in \mathbb{R}^{n \times r}$ and weighting matrix $\mathbf{G} \in \mathbb{R}^{r \times \ell T}$, we could solve the following optimization problem:

$$\min_{\mathbf{U}, \mathbf{G}} \quad \frac{1}{2} \|\mathbf{Y} - \mathcal{A}(\mathbf{UG})\|_{\mathsf{F}}^2, \quad \text{where} \quad \mathbf{Y} := [\mathbf{Y}_1 \ \ldots \ \mathbf{Y}_T] \in \mathbb{R}^{m \times \ell T}, \tag{2}$$

and $\mathcal{A}(\cdot) : \mathbb{R}^{n \times \ell T} \to \mathbb{R}^{m \times \ell T}$ is a linear sensing operator. We refer to this approach and related techniques as "static" methods, as they do not account for any temporal component of the underlying data. This problem has been widely studied in the literature and can be solved using various forms of alternating gradient-based methods (Jain et al., 2013). However, for applications with time-varying data, static methods may be suboptimal, as they ignore the temporal structure that could potentially be used to make more accurate data estimates.

For settings in which $\mathcal{A}(\cdot)$ is the identity map, there are many classical works on subspace tracking that approach the objective in Equation (2) with data streaming over time, but use constant step-sizes or linear-algebraic methods that essentially allow tracking of time-varying subspaces (Yang, 1995; Haghighatshoar & Caire, 2018; Narayanamurthy & Vaswani, 2018b; Vaswani et al., 2018; Comon & Golub, 1990). The theoretical results in these earlier works are generally limited to asymptotic convergence guarantees with static underlying subspaces. On the other hand, Narayanamurthy & Vaswani (2018b) relax the assumption of constant subspace to a very slowly varying subspace between the change points. For a review of these methods, see Vaswani et al. (2018). Dynamic subspace estimation has also been studied in the batch setting where one has access to all the data at once, such as (Saad-Falcon et al., 2024a) and the following papers that use the Grassmannian geodesic model discussed in this paper (Lake & Keenan, 1998b; Fuhrmann, 1997; Srivastava & Klassen, 2004; Hong et al., 2016).

When data are observed through compressive maps $\mathcal{A}(\cdot)$, such as missing data or general compressive operators, the existing works are those that build on the subspace tracking algorithms based on optimization techniques. For example, the PETRELS algorithm (Chi et al., 2012) is a recursive least-squares approach for the missing data setting, and provides convergence theory that assumes that the subspace changes at a particular instant and then stays constant for sufficient time so that the change can be tracked (also called the piecewise constant model). Oja's method and GROUSE (Allen-Zhu & Li, 2017; Balzano et al., 2010) can also handle missing data or compressive observations (Zhang & Balzano, 2016; Balzano, 2022). However, the main difference between our algorithm and methods like GROUSE is that while they use geodesics (e.g., in their stochastic gradient updates), the geodesic is primarily employed as an optimization tool: each update follows a different geodesic determined by the current gradient direction to optimize a fixed objective in a streaming or online manner. This streaming aspect is often used as a heuristic to handle slowly time-varying data. In contrast, as we present in the following section, we model the time-varying subspace itself as a geodesic curve on the Grassmannian. For a more comprehensive survey of missing data tracking methods, see Balzano et al. (2018a).

While we defer a comparison of our work with these existing methods in the case where the linear operator is the identity mapping to Appendix B (see Figure 12), we primarily compare our approach to the static method in Equation (2). This choice is motivated by two reasons: (i) it is often nontrivial to extend these methods to cases where the linear operator is not the identity, and (ii) many existing works assume that subspaces change one at a time and that the changes between subspaces are minimal—assumptions that are not made by our algorithm.

### 2.3 Proposed Method: A Dynamic Approach via Grassmannian Geodesics

In this work, we consider estimating all data points $\{\mathbf{x}_{i,j}\}$ generated by the sequence of subspaces $\{\mathbf{U}_i\}$ by modeling the subspaces as a geodesic on the Grassmann manifold. Similar to lines in Euclidean space, Grassmannian geodesics behave as interpolating curves, and can be expressed as (Absil et al., 2004, Section 3.8):

$$\mathbf{U}_i := \mathbf{U}(t_i) = \mathbf{H}\cos(\mathbf{\Theta}t_i) + \mathbf{Z}\sin(\mathbf{\Theta}t_i), \tag{3}$$

where $\mathbf{H} \in \mathcal{V}^{n\times r}$ is an orthonormal basis for a point on the Grassmannian, $\mathbf{Z} \in \mathcal{V}^{n\times r}$ is a matrix with orthonormal columns whose span is in the tangent space of range($\mathbf{H}$) (i.e., $\{\mathbf{Z} \in \mathcal{V}^{n\times r} \mid \mathbf{H}^\top \mathbf{Z} = \mathbf{0}\}$), $\mathbf{\Theta} \in \mathbb{R}^{r\times r}$ is a diagonal matrix with entries $\theta_s$, where $\theta_s$ denotes the principal angle between two endpoints of a geodesic, and $t_i \in [0,1]$ is a scalar that represents the time of each $\mathbf{U}_i$.

Geometrically, we can think of $\mathbf{H}$ as some starting point on the Grassmannian, $\mathbf{Z}$ as a normalized direction we want to travel, and $\mathbf{\Theta}t_i$ as the distance to reach $\mathbf{U}_i$. If we are given the time points $t_i$ (which we treat as tunable hyperparameters), then $\mathbf{H}, \mathbf{Z}$, and $\mathbf{\Theta}$ are all learnable parameters of $\mathbf{U}_i$ that we can use to estimate $\{\mathbf{x}_{i,j}\}$. Then, instead of using a hard constraint $\mathbf{x}_{i,j} = \mathbf{U}_i\mathbf{g}_{i,j}$ as in Equation (2), we propose using a regularizer that penalizes data points that lie outside of their respective subspaces. By collecting all learnable parameters into $\mathbf{\Psi} = \{\mathbf{H}, \mathbf{Z}, \mathbf{\Theta}\} \cup \{\mathbf{x}_{i,j}\}$, we arrive at the following optimization problem that forms the central component of this work:

$$\min_{\mathbf{\Psi}} \quad f(\mathbf{\Psi}) := \sum_{i=1}^{T}\sum_{j=1}^{\ell} \underbrace{\tfrac{1}{2}\|\mathbf{y}_{i,j} - \mathbf{A}_{i,j}\mathbf{x}_{i,j}\|_2^2}_{\text{data consistency}} + \underbrace{\tfrac{\lambda}{2}\|(\mathbf{I}_n - \mathbf{U}_i\mathbf{U}_i^\top)\mathbf{x}_{i,j}\|_2^2}_{\text{subspace regularizer}}$$

$$\text{s.t.} \quad \underbrace{\mathbf{U}_i = \mathbf{H}\cos(\mathbf{\Theta}t_i) + \mathbf{Z}\sin(\mathbf{\Theta}t_i)}_{\text{constrain to Grassmannian geodesic}}. \tag{4}$$

Throughout, $f$ refers to this cost function, which for simplicity we often write compactly as

$$\min_{\mathbf{\Psi}} \quad f(\mathbf{\Psi}) = \sum_{i=1}^{T}\sum_{j=1}^{\ell} \frac{1}{2}\big\|\widetilde{\mathbf{y}}_{i,j} - \widetilde{\mathbf{A}}_{i,j}\mathbf{x}_{i,j}\big\|_2^2$$

$$\text{s.t.} \quad \mathbf{U}_i = \mathbf{H}\cos(\mathbf{\Theta}t_i) + \mathbf{Z}\sin(\mathbf{\Theta}t_i), \quad \forall i \in [T], \tag{5}$$

where we define

$$\widetilde{\mathbf{y}}_{i,j} := \begin{bmatrix}\mathbf{y}_{i,j}\\ \mathbf{0}\end{bmatrix}, \qquad \widetilde{\mathbf{A}}_{i,j} := \begin{bmatrix}\mathbf{A}_{i,j}\\ \sqrt{\lambda}\,(\mathbf{I}_n - \mathbf{U}_i\mathbf{U}_i^\top)\end{bmatrix}.$$

**Remarks.** Here, we provide an argument for why this regularization will provide a well-posed optimization problem, despite having very few linear measurements of each vector. Define $\mathbf{B}_i := \mathbf{I}_n - \mathbf{U}_i\mathbf{U}_i^\top$ and

$$\mathbb{A}_{i,j} := \text{range}(\mathbf{A}_{i,j}^\top), \qquad \mathbb{B}_i := \text{range}(\mathbf{B}_i^\top).$$

If each of the measurement matrices in $\{\mathbf{A}_{i,j}\}$ has rank $m \geq r$ and $\dim(\mathbb{A}_{i,j} \cap \mathbb{B}_i) \leq m - r$, then we can lower bound the rank of $\widetilde{\mathbf{A}}_{i,j} \in \mathbb{R}^{(m+n)\times n}$ by

$$\text{rank}\left(\widetilde{\mathbf{A}}_{i,j}\right) = \text{rank}\left(\widetilde{\mathbf{A}}_{i,j}^\top\right) = \dim(\mathbb{A}_{i,j}) + \dim(\mathbb{B}_i) - \dim(\mathbb{A}_{i,j} \cap \mathbb{B}_i)$$

$$\geq m + (n - r) - (m - r) = n.$$

Hence, $\widetilde{\mathbf{A}}_{i,j}$ would have full column rank and $\{\mathbf{x}_{i,j}\}$ would be identifiable, assuming we knew the subspaces $\{\mathbf{U}_i\}$. The assumption on the intersection between two subspaces is natural in a random, high-dimensional setting – if $\{\mathbf{A}_{i,j}\}$ are random with a continuous distribution over $\mathbb{R}^{m\times n}$ that were independent of $\{\mathbf{U}_i\}$, then $\dim(\mathbb{A}_{i,j} \cap \mathbb{B}_i) = m - r$.

Of course, since we do not know the subspaces $\{\mathbf{U}_i\}$ a priori, we need to estimate them from the data. Without the geodesic constraint, the problem would amount to estimating a total of $T$ unrelated subspaces.

For applications with time-varying data, it is often favorable to impose a relationship between consecutive subspaces $\mathbf{U}_i$. The Grassmannian geodesic constraint serves as a natural remedy for these problems, as we only need to estimate a $2r$-dimensional basis represented by $\mathbf{H} \in \mathcal{V}^{n \times r}$ and $\mathbf{Z} \in \mathcal{V}^{n \times r}$ across *all* time points, while enforcing smoothness between consecutive subspaces. Consequently, another advantage of this constraint is that we can allow $\ell < r$ (i.e., observe fewer data points than the intrinsic rank of the subspace) and still achieve accurate recovery.

## 3  Riemannian Block Majorize-Minimize Algorithm

This section presents a Riemannian block majorize-minimization (RBMM) algorithm for solving Equation (4). The algorithm updates one block of $\boldsymbol{\Psi}$ at a time while holding the others fixed: each update minimizes a surrogate that majorizes the marginal of $f$ in the active block, which we denote $f_{(p)}$ for the $p$-th block; when the active block is clear from the context, we equivalently write the marginal as its argument (e.g., $f(\mathbf{x}_{i,j})$). Section 3.1 presents the update steps for the basis $\mathbf{Q} \coloneqq [\mathbf{H}, \mathbf{Z}] \in \mathcal{V}^{n \times 2r}$, each diagonal coordinate of $\boldsymbol{\Theta} \in \mathbb{R}^{r \times r}$, and each data sample $\mathbf{x}_{i,j} \in \mathbb{R}^n$, and Section 3.2 introduces a spectral scheme for initializing each factor. Algorithm 1 summarizes the procedure. For our theoretical analysis, we introduce proximal parameters $\lambda_{\mathbf{Q}}$, $\lambda_{\boldsymbol{\Theta}}$, $\lambda_{\mathbf{x}}$, one per block, to ensure convergence to a stationary point; however, our experiments show that they do not play a significant role in performance and need not be tuned.

### 3.1  Riemannian Block Update Steps

This section summarizes the update steps for our RBMM algorithm. Appendix A derives the complete update steps. Throughout this section, for simplicity, we denote

$$\mathbf{U}_i = \mathbf{H}\cos(\boldsymbol{\Theta}t_i) + \mathbf{Z}\sin(\boldsymbol{\Theta}t_i) = \mathbf{Q}\mathbf{R}_i, \quad \text{where} \quad \mathbf{Q} \coloneqq [\mathbf{H}, \mathbf{Z}], \quad \mathbf{R}_i \coloneqq \begin{bmatrix} \cos(\boldsymbol{\Theta}t_i) \\ \sin(\boldsymbol{\Theta}t_i) \end{bmatrix}.$$

**Updates for X.** Suppose $\mathbf{U}_i = \mathbf{Q}\mathbf{R}_i$ are fixed and define $f(\mathbf{x}_{i,j}) = \frac{1}{2}\|\widetilde{\mathbf{y}}_{i,j} - \widetilde{\mathbf{A}}_{i,j}\mathbf{x}_{i,j}\|_2^2$. We construct a majorizer $g^k$ at iteration $k$ for $f$ from the second-order Taylor expansion and then update $\mathbf{X}$ by minimizing that majorizer:

$$\mathbf{x}_{i,j}^{k+1} = \underset{\mathbf{x}_{i,j}}{\arg\min}\, g^k(\mathbf{x}_{i,j}; \mathbf{x}_{i,j}^k) = \underset{\mathbf{x}_{i,j}}{\arg\min}\, \left\| \mathbf{x}_{i,j} - \left( \mathbf{x}_{i,j}^k - \frac{1}{L + \lambda_{\mathbf{x}}}\nabla f(\mathbf{x}_{i,j}^k) \right) \right\|_2^2 = \mathbf{x}_{i,j}^k - \frac{1}{L + \lambda_{\mathbf{x}}}\nabla_{\mathbf{x}} f(\mathbf{x}_{i,j}^k), \quad (6)$$

where $L$ is the Lipschitz constant of $f$, $\nabla_{\mathbf{x}} f(\mathbf{x}_{i,j}^k) = \widetilde{\mathbf{A}}_{i,j}^\top(\widetilde{\mathbf{A}}_{i,j}\mathbf{x}_{i,j}^k - \widetilde{\mathbf{y}}_{i,j})$ and $\lambda_{\mathbf{x}} > 0$ is the parameter introduced by adding a proximal term to ensure a quadratic majorization gap (see Appendix A). This update step is a gradient descent step with a fixed step size that ensures a monotonically non-increasing objective function value.

**Updates for Q.** To update $\mathbf{Q}$, suppose that each $\mathbf{R}_i$ and $\mathbf{X}_i$ are fixed. Then, notice that the data consistency term in Equation (4) is not a function of $\mathbf{Q}$, so we only need to consider the subspace regularizer:

$$\widehat{\mathbf{Q}} = \underset{\mathbf{Q} \in \mathcal{V}^{n \times 2r}}{\arg\min}\, f(\mathbf{Q}) = \underset{\mathbf{Q} \in \mathcal{V}^{n \times 2r}}{\arg\min}\, \frac{\lambda}{2}\sum_{i=1}^T \sum_{j=1}^\ell \left\| \left(\mathbf{Q}\mathbf{R}_i\mathbf{R}_i^\top\mathbf{Q}^\top - \mathbf{I}_n\right)\mathbf{x}_{i,j} \right\|_2^2 \tag{7}$$

$$= \underset{\mathbf{Q} \in \mathcal{V}^{n \times 2r}}{\arg\min}\, \frac{\lambda}{2}\sum_{i=1}^T \left\| \left(\mathbf{Q}\mathbf{R}_i\mathbf{R}_i^\top\mathbf{Q}^\top - \mathbf{I}_n\right)\mathbf{X}_i \right\|_{\mathsf{F}}^2. \tag{8}$$

By constructing a linear majorizer $g^k$ and projecting onto the Stiefel manifold (Absil et al., 2007; Higham, 1989), we obtain

$$\mathbf{Q}^{k+1} = \underset{\mathbf{Q} \in \mathcal{V}^{n \times 2r}}{\arg\min}\, g^k(\mathbf{Q}; \mathbf{Q}^k) = \left\| \mathbf{Q} - \left(\lambda_{\mathbf{Q}}\mathbf{Q}^k - \nabla f_{\mathbf{Q}}(\mathbf{Q}^k)\right) \right\|_{\mathsf{F}}^2 \tag{9}$$

$$= \mathbf{W}\mathbf{V}^\top, \quad \text{where} \quad \left(\lambda_{\mathbf{Q}}\mathbf{Q}^k - \nabla f_{\mathbf{Q}}(\mathbf{Q}^k)\right) = \mathbf{W}\boldsymbol{\Sigma}\mathbf{V}^\top, \tag{10}$$

and $\nabla f_{\mathbf{Q}}(\mathbf{Q}) = -\lambda\sum_{i=1}^T \mathbf{X}_i\mathbf{X}_i^\top\mathbf{Q}\mathbf{R}_i\mathbf{R}_i^\top$ and $\lambda_{\mathbf{Q}} > 0$ is the proximal parameter for $\mathbf{Q}$.

**Updates for $\boldsymbol{\Theta}$.** To preserve the diagonal structure of $\boldsymbol{\Theta}$, we update each coordinate of $\boldsymbol{\Theta}$ present in each $\mathbf{R}_i$, which we denote as $\theta_s \in \mathbb{R}$, $\forall s \in [r]$. Similarly, consider the objective function

$$\widehat{\boldsymbol{\Theta}} = \arg\min_{\boldsymbol{\Theta}} f(\boldsymbol{\Theta}) = \arg\min_{\boldsymbol{\Theta}} \frac{\lambda}{2} \sum_{i=1}^{T} \left\| \left( \mathbf{Q}\mathbf{R}_i\mathbf{R}_i^\top\mathbf{Q}^\top - \mathbf{I}_n \right) \mathbf{X}_i \right\|_{\mathsf{F}}^2.$$

Simplifying this loss function in terms of $\theta_s$ leads to

$$\widehat{\theta}_s = \arg\min_{\theta_s} -\frac{\lambda}{2} \sum_{i=1}^{T} \sum_{s=1}^{r} f_{i,s}(\theta_s) = \arg\min_{\theta_s} -\frac{\lambda}{2} \sum_{i=1}^{T} \sum_{s=1}^{r} r_{i,s} \cdot \cos(2\theta_s t_i - \phi_{i,s}), \tag{11}$$

with the following definitions:

$$\alpha_{i,s} := \left[\mathbf{H}^\top\mathbf{X}_i\mathbf{X}_i^\top\mathbf{H}\right]_{s,s} \qquad\qquad r_{i,s} := \sqrt{\left(\frac{\alpha_{i,s} - \gamma_{i,s}}{2}\right)^2 + \beta_{i,s}^2}$$

$$\beta_{i,s} := \mathrm{real}\left\{\left[\mathbf{Z}^\top\mathbf{X}_i\mathbf{X}_i^\top\mathbf{H}\right]_{s,s}\right\} \qquad \phi_{i,s} := \mathrm{arctan2}\left(\beta_{i,s}, \frac{\alpha_{i,s} - \gamma_{i,s}}{2}\right)$$

$$\gamma_{i,s} := \left[\mathbf{Z}^\top\mathbf{X}_i\mathbf{X}_i^\top\mathbf{Z}\right]_{s,s}. \tag{12}$$

Upon constructing a quadratic majorizer (Funai et al., 2008) (see Appendix A.3), we have the update step

$$\theta_s^{k+1} = \theta_s^k - \sum_{i=1}^{T} \frac{\dot{f}_{i,s}(\theta_s^k)}{w_{f_{i,s}}(\theta_s^k)}, \tag{13}$$

where $\dot{f}_{i,s}(\theta_s^k) = \lambda r_{i,s} t_i \sin(2\theta_s t_i - \phi_{i,s})$ and

$$w_{f_{i,s}}(\theta_s^k) = \begin{cases} \dfrac{\dot{f}_{i,s}(\theta_s^k)}{\mathrm{mod}\left(\left(\theta_s^k - \frac{\phi_{i,s}}{2t_i}\right) + \frac{\pi}{2t_i}, \frac{2\pi}{2t_i}\right) - \frac{\pi}{2t_i}} + \lambda_{\boldsymbol{\Theta}} & \theta_s^k \neq \frac{\phi_{i,s}+2\pi h}{2t_i}, \ h \in \mathbb{Z} \\ 2\lambda r_{i,s} t_i^2 + \lambda_{\boldsymbol{\Theta}} & \theta_s^k = \frac{\phi_{i,s}+2\pi h}{2t_i}, \ h \in \mathbb{Z}, \end{cases} \tag{14}$$

where $\lambda_{\boldsymbol{\Theta}} > 0$ is the proximal parameter for $\boldsymbol{\Theta}$.

### 3.2 Spectral Initialization

Due to the non-convex nature of our problem, a smart initialization facilitates convergence to good solutions. This section presents a spectral initialization scheme for each of the blocks that empirically improves the performance of our algorithm.

**Initialization for $\mathbf{X}$.** Suppose that every entry of the sensing matrix $\mathbf{A}_{i,j}$ had an independent, random distribution over $\mathbb{R}^{m \times n}$ with zero mean and unit variance. To initialize each $\mathbf{x}_{i,j}$, consider the surrogate matrix

$$\mathbf{C}_{i,j} = \mathbf{B}_{i,j}\mathbf{A}_{i,j}, \quad \text{where } \mathbf{B}_{i,j} = \mathrm{Diag}(\mathbf{y}_{i,j}). \tag{15}$$

After some elementary calculations, we have

$$\mathbb{E}\left[\mathbf{C}_{i,j}^\top\right] = \begin{bmatrix} \mathbf{x}_{i,j} & \mathbf{x}_{i,j} & \dots & \mathbf{x}_{i,j} \end{bmatrix},$$

which is a rank-1 matrix, and so we initialize each $\mathbf{x}_{i,j}$ by taking the first left singular vector of $\mathbf{C}_{i,j}$.

**Initialization for $\mathbf{H}, \mathbf{Z}, \boldsymbol{\Theta}$.** We first introduce a *random* scheme for initializing the geodesic parameters. Let $\mathbf{H}_1, \mathbf{H}_2 \in \mathcal{V}^{n \times r}$ be two orthonormal bases for two points on the Grassmannian. Given $t_i \in [0, 1]$, a point in between these two endpoints is given

$$\mathbf{U}_i = \mathrm{span}\left(\mathbf{H}_1\mathbf{V}\cos(\boldsymbol{\Theta}t_i) + \mathbf{Z}\sin(\boldsymbol{\Theta}t_i)\right), \tag{16}$$

---

**Algorithm 1** RBMM Algorithm for Dynamic Subspace Estimation

---

**Require:** Observations $\{\mathbf{y}_{i,j}\} \in \mathbb{R}^m$; Sensing matrices $\{\mathbf{A}_{i,j}\} \in \mathbb{R}^{m \times n}$; Parameters $\lambda_{\mathbf{x}}, \lambda_{\mathbf{Q}}, \lambda_{\mathbf{\Theta}}, \lambda \geq 0$; Iterations $K$; Geodesic parameters $t_i \in [0,1], \forall i \in [T]$

  1: **for** $i = 1, \ldots, T$ **do**
  2:     **for** $j = 1, \ldots, \ell$ **do**
  3:         $\mathbf{C}_{i,j} = \mathbf{B}_{i,j}\mathbf{A}_{i,j}$                         ▷ Construct surrogate matrix in Equation (15)
  4:         $\mathbf{x}_{i,j}^0 \leftarrow \text{SVD}(\mathbf{C}_{i,j})$                   ▷ Initialize via top left singular vector of $\mathbf{C}_{i,j}$
  5:     **end for**
  6: **end for**
  7: $\mathbf{X}_i^0 = [\mathbf{x}_{i,1}^0 \ldots \mathbf{x}_{i,\ell}^0] \in \mathbb{R}^{n \times \ell}$              ▷ Concatenate initialized samples at time point $i$
  8: $\mathbf{X}^0 = [\mathbf{X}_1^0 \ldots \mathbf{X}_T^0] \in \mathbb{R}^{n \times \ell T}$              ▷ Concatenate all initial samples across time
  9: $\mathbf{H}^0, \mathbf{Z}^0, \mathbf{\Theta}^0 \leftarrow \text{SVD}(\mathbf{X}^0)$                     ▷ Initialize via Equations (16, 17)
10: **for** $k = 1, \ldots, K$ **do**                                ▷ Start RBMM algorithm
11:     **for** $i = 1, \ldots, T$ **do**
12:         **for** $j = 1, \ldots, \ell$ **do**
13:             $\mathbf{x}_{i,j}^{k+1} = \mathbf{x}_{i,j}^k - \frac{1}{L+\lambda_{\mathbf{x}}}\nabla_{\mathbf{x}}f(\mathbf{x}_{i,j}^k)$         ▷ Update signals $\mathbf{x}_{i,j}$ from Equation (6)
14:         **end for**
15:     **end for**
16:     $\left(\lambda_{\mathbf{Q}}\mathbf{Q}^k - \nabla f_{\mathbf{Q}}(\mathbf{Q}^k)\right) = \mathbf{W}\mathbf{\Sigma}\mathbf{V}^\top$                         ▷ Compute SVD
17:     $\mathbf{Q}^{k+1} = \mathbf{W}\mathbf{V}^\top$                            ▷ Update $\mathbf{Q}$ from Equation (9)
18:     **for** $s = 1, \ldots, r$ **do**
19:         $\theta_s^{k+1} = \theta_s^k - \sum_{i=1}^T \frac{\dot{f}_{i,s}(\theta_s^k)}{w_{f_{i,s}}(\theta_s^k)}$         ▷ Update each $\theta_s$ from Equations (13, 14)
20:     **end for**
21: **end for**
22: **Return:** $\mathbf{X}_i^K, \ \forall i \in [T]$                               ▷ Return estimated data points

---

where $\left(\mathbf{I} - \mathbf{H}_1\mathbf{H}_1^\top\right)\mathbf{H}_2\left(\mathbf{H}_1^\top\mathbf{H}_2\right)^{-1} = \mathbf{Z}\mathbf{S}\mathbf{V}^\top$ is a compact SVD and $\mathbf{\Theta} = \tan^{-1}(\mathbf{S})$. Thus, a random initialization for $\mathbf{H} \in \mathcal{V}^{n \times r}, \mathbf{Z} \in \mathcal{V}^{n \times r}$ and $\mathbf{\Theta} \in \mathbb{R}^{r \times r}$ takes any two orthonormal bases and applies Equation (16).

However, we can obtain more accurate initial points by using information from our data. Suppose $n \geq 2r$ and that we observed the data $\mathbf{X}$ directly (i.e., $\mathbf{A}_{i,j} = \mathbf{I}_n$). If we assumed that $\mathbf{X}$ was truly generated from a geodesic, based on Figure 1, we can expect the two endpoints of the geodesic, $\mathbf{U}_1$ and $\mathbf{U}_T$, to serve as bases for two endpoints of the geodesic. Then, we can take two subsets of the concatenated data as follows:

$$\mathbf{X} = [\underbrace{\mathbf{X}_1 \overbrace{\mathbf{x}_{2,1} \ldots \mathbf{x}_{2,\ell}}^{\mathbf{X}_2}}_{=\mathbf{D}_1 \in \mathbb{R}^{n \times r}} \ldots \overbrace{\mathbf{x}_{T-1,1} \ldots \underbrace{\mathbf{x}_{T-1,\ell}}_{=\mathbf{D}_2 \in \mathbb{R}^{n \times r}} \mathbf{X}_T}^{\mathbf{X}_{T-1}}]. \tag{17}$$

We can then use the left singular vectors of $\mathbf{D}_1$ and $\mathbf{D}_2$ as $\mathbf{H}_1$ and $\mathbf{H}_2$, respectively, and apply Equation (16). We call this procedure *spectral* initialization. Since we do not observe $\mathbf{X}$ directly, we use the initialization method for $\mathbf{X}$ above and apply spectral initialization for the geodesic parameters. Note that Equation (17) constructs $\mathbf{D}_1$ using $\mathbf{X}_1$ and $\mathbf{x}_{2,1}$ only for illustration in the case $\ell = r - 1$. When $\ell \geq r$, one can instead use only $\mathbf{X}_1$; when $\ell < r - 1$, one can use additional samples from $\mathbf{X}_t$ for $t > 1$.

## 4 THEORETICAL RESULTS

This section presents the theoretical guarantees of our RBMM algorithm.

**Theorem 1** (Convergence to a Stationary Point)**.** *Let* $\mathbf{\Psi}^k = \{\mathbf{X}_i^k\}_{i=1}^T \cup \{\mathbf{Q}^k, \mathbf{\Theta}^k\}$ *denote the iterates generated by Algorithm 1 at iteration $k$ starting from any initialization with proximal parameters* $\lambda_{\mathbf{x}}, \lambda_{\mathbf{Q}}, \lambda_{\mathbf{\Theta}} > 0$. *For any $\epsilon > 0$, if $k = \widetilde{\mathcal{O}}(\epsilon^{-2})$, then we have*

$$\sum_{p=1}^{T+2} \|\nabla f_{(p)}(\mathbf{\Psi}_p^k)\| \leq \epsilon, \tag{18}$$

| Measurements | Reference | $r$-SVD | $2r$-SVD | $r$-Geodesic |
|---|---|---|---|---|

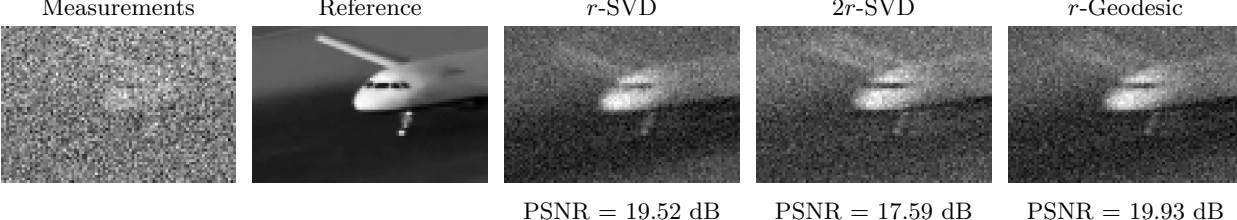

| | | PSNR = 19.52 dB | PSNR = 17.59 dB | PSNR = 19.93 dB |

Figure 2: Visual results with corresponding PSNRs for denoising the plane video dataset with Gaussian noise with standard deviation $\sigma_{\mathbf{y}} = 0.4$ and rank $r = 1$.

where $p$ indexes the $T + 2$ sequentially updated blocks of variables in $\boldsymbol{\Psi}^k$.

Appendix C provides all proofs. The proximal parameters in Theorem 1 can be viewed as values that ensure each update step is bounded in some sense – values closer to zero result in larger update steps, while larger values restrict the update steps, consequently leading to slower convergence. Due to the non-convex nature of our problem, it is very challenging to obtain guarantees of convergence to the global minimum. However, from our synthetic experiments in a planted geodesic setting, we observe empirically that our method with spectral initialization often converges to the ground truth (see Figure 5).

**Theorem 2** (Monotonic Objective Function). *Let $\boldsymbol{\Psi}^k = \{\mathbf{X}_i^k\}_{i=1}^T \cup \{\mathbf{Q}^k, \boldsymbol{\Theta}^k\}$ denote the iterates generated by Algorithm 1 at iteration $k$. Algorithm 1 produces blocks that are monotonically non-increasing in the loss: $f(\boldsymbol{\Psi}^{k+1}) \leq f(\boldsymbol{\Psi}^k)$.*

Generally, existing algorithms that use optimization methods such as gradient descent do not guarantee a monotonically non-increasing objective function unless the step size is appropriately tuned. Our algorithm possesses this property without requiring additional step size tuning, which is particularly advantageous in domains with intrinsically high-dimensional data, where cross-validation can be costly. Theorem 2 holds independently of the proximal parameters – they can be set to any non-negative value and the statement will remain valid.

## 5 EXPERIMENTS

This section presents our experimental results that are divided into two parts: experiments with fully sampled data (Section 5.1) and with undersampled data (Section 5.2). Both parts include experiments on synthetic and real video data, aimed at analyzing the behavior and performance of our proposed algorithm. For our algorithm, we set $\lambda_{\mathbf{x}} = \lambda_{\mathbf{Q}} = \lambda_{\boldsymbol{\Theta}} = 0$ and $\lambda = 1$, unless otherwise specified. We considered the three following performance metrics:

$$\text{Subspace Error} = \frac{1}{2rT} \sum_{i=1}^{T} \|\mathbf{U}_i \mathbf{U}_i^\top - \widehat{\mathbf{U}}_i \widehat{\mathbf{U}}_i^\top\|_{\mathsf{F}}^2,$$

for any two subspaces $\mathbf{U}_i$ and $\widehat{\mathbf{U}}_i$, the normalized root mean-squared error (NRMSE), defined by $\|\widehat{\mathbf{X}} - \mathbf{X}\|_{\mathsf{F}}/\|\mathbf{X}\|_{\mathsf{F}}$, and the peak-signal-to-noise ratio (PSNR).

### 5.1 Experiments with Fully Sampled Data

Before presenting experimental results for the undersampled data case, we first consider the fully sampled setting, i.e., $\mathbf{A}_{i,j} = \mathbf{I}_n$. In this case, the data consistency term in Equation (4) simplifies to a denoising task, where the data points are regularized to lie on a geodesic. The goal of this setting is to (i) understand the behavior of the RBMM algorithm and how it differs from the undersampled case, and (ii) investigate how enforcing a geodesic structure aids in denoising compared to static approaches, i.e., the truncated SVD.

**Synthetic Experiments.** Here, we performed experiments in a planted geodesic setting to investigate the dependence of the algorithm's performance on its parameters. To this end, we generated measurements

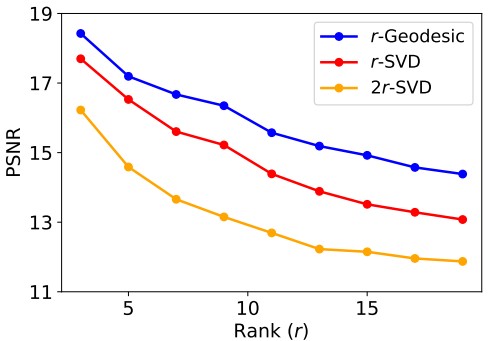
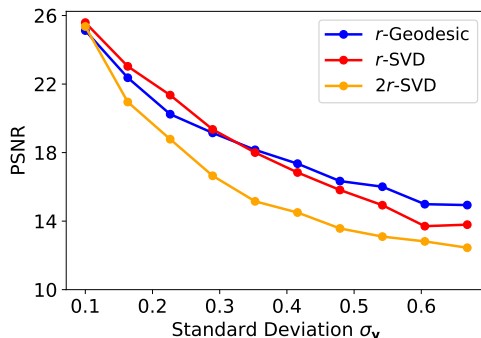

Figure 3: Experimental results on the plane video dataset for denoising with fully observed data. Left: Plot of the PSNR for varying values of the rank $r$ for fixed $\sigma_{\mathbf{y}} = 0.40$. Right: Plot of the PSNR for varying values of the standard deviation of the added noise $\sigma_{\mathbf{y}}$ for a fixed rank $r = 5$. The PSNR is averaged across all video frames.

according to

$$\mathbf{y}_{i,j} = \mathbf{U}_i \mathbf{g}_{i,j} + \boldsymbol{\eta}_{i,j} \in \mathbb{R}^n, \qquad \forall j \in [\ell], \ \forall i \in [T], \tag{19}$$

where $\mathbf{g}_{i,j} \sim \mathcal{N}(0, \mathbf{I}_r)$ and $\boldsymbol{\eta}_{i,j} \sim \mathcal{N}(0, \sigma_{\mathbf{y}} \cdot \mathbf{I}_m)$, with $n = 40$. Figure 4 plots the subspace error for our algorithm and the $r$-SVD for different values of $\ell$, where $T = 51$, $r = 4$, and $\sigma_{\mathbf{y}} = 10^{-2}$. The SVD method involves computing the top-$r$ SVD components of the data at each time point and, hence, can only be computed when $\ell \geq r$. On the other hand, our algorithm works even for $\ell < r$ while achieving a lower error.

Figure 5a shows the fraction of times our algorithm converged to the ground-truth solution as measured by the subspace error, for $\ell = 1$, across varying values of $T$ and $r$. There is a clear phase transition at $T = 2r$, as the geodesic assumption amounts to estimating $2nr$ parameters from a total of $Tn\ell$ data points.

**Video Data Experiments.** Next, we present results on a video denoising task, where the video consists of frames showing a plane landing on a runway. The video has a total of 105 frames, each with dimensions $(60 \times 90)$. This video was used in other inverse problem contexts (Nayer et al., 2020; Kwon et al., 2022), albeit for a different application. For preprocessing, we normalized pixel values to $[0, 1]$ and mean-centered the data

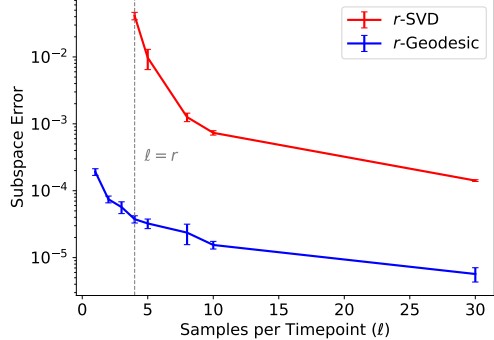

Figure 4: Subspace error for varying $\ell$ in the fully sampled case.

after vectorizing each frame. For our algorithm, we chose $r = 5$ and set $\ell = r - 1$ unless otherwise specified. This case emphasizes the fact that our algorithm can handle $\ell < r$. We evenly sampled the time steps $t_i \in [0.15, 0.85]$, as this range generally yielded the best performance. We compared our algorithm against the $r$-SVD and $2r$-SVD baselines, which performed an SVD of the corresponding rank on the data and projected the frames onto the resulting subspaces. We consider $2r$ as a baseline, as $\mathbf{Q}$ spans a $2r$-dimensional subspace.

Figure 2 presents visual results on a single frame, demonstrating that our algorithm denoised the video more effectively than the SVD-based baselines in terms of PSNR when $\sigma_{\mathbf{y}} = 0.40$. This suggests that incorporating additional structure can be beneficial for such tasks, especially in highly noisy settings. Figure 3 shows the PSNR for varying values of the rank for a fixed $\ell = 4$ (left) and the standard deviation of the added noise (right). In less noisy settings, the SVD-based solutions appear to outperform our method, but the gap quickly disappears as the noise level increases.

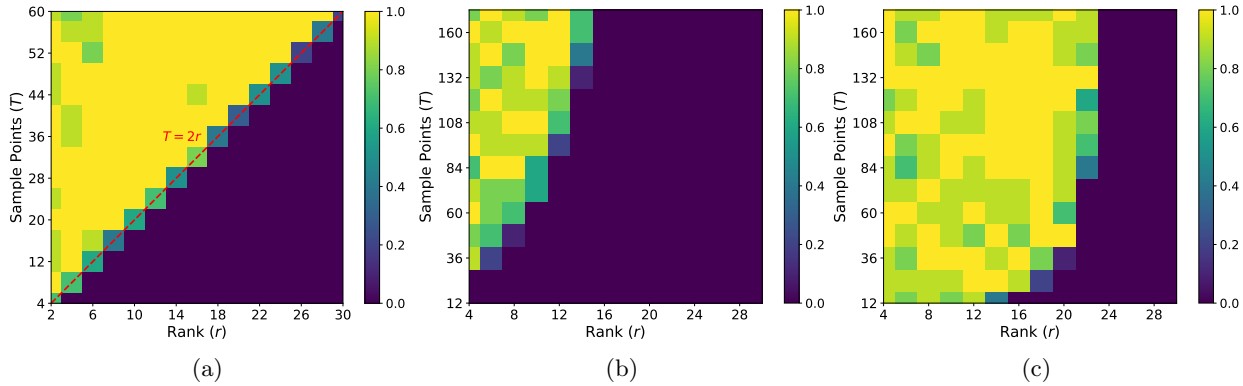

Figure 5: Fraction of convergence to ground truth for varying values of $r$ and $T$ over 10 random runs for a fixed $m$. The fraction is the number of times the subspace error went below a threshold of $10^{-4}$ over the total number of trials. (a): Fully-sampled case with $m = n = 40$ and $\ell = 1$. (b): Under-sampled case with $m = \frac{1}{2}n = 20$ and $\ell = 1$. (c): Under-sampled case with $m = \frac{1}{2}n = 20$ and $\ell = r - 1$. For the under-sampled settings, the number of data points $\ell$ for each time point makes a notable difference in the recovery of the underlying subspaces.

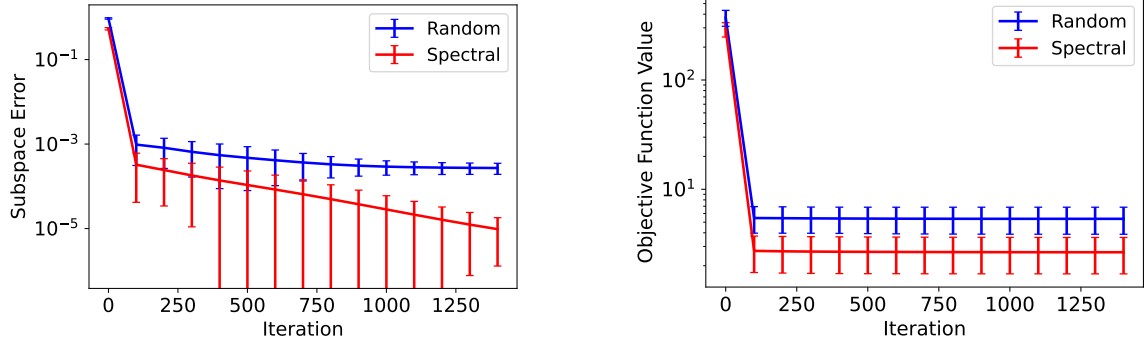

Figure 6: Results showing the benefits of spectral initialization with $\sigma_{\mathbf{y}} = 0.01$ averaged over 10 runs for matrix completion. The spectral initialization scheme leads to faster convergence in terms of iterations.

## 5.2 Experiments with Undersampled Data

This section presents experimental results in the under-sampled setting. Similar to Section 5.1, we experimented with both synthetic and video data, and we also include experiments on real data, which involve reconstructing fMRI images from sub-sampled $k$-space data. In the under-sampled setting, since we do not have access to the full dataset, we selected the rank for the video data through trial and error, and we present results using two different rank choices. For the fMRI data, we chose the rank based on a data-sharing method, which involves searching for the nearest row of the non-missing data sample and using that to reconstruct the data. In Figure 19a, we show that the singular value plots for the data-shared reconstructed frames and the original frames exhibit similar patterns, justifying the choice of rank.

**Synthetic Experiments.** Similar to Section 5.1, we considered experiments in a planted geodesic setting, where the objective is to: (i) demonstrate the efficiency of our spectral initialization method compared to random initialization; (ii) observe the dependency of the parameters and their effects on the performance in contrast to the fully-sampled setting; (iii) compare the performance of our algorithm on noisy matrix sensing to the baseline AltMin (Jain et al., 2013), which is an alternating minimization scheme for updating two $r$-dimensional factors. Since AltMin is a matrix recovery algorithm, we vectorized and stacked all of the signals to construct a single matrix. Appendix B.2 provides more experimental details on the baselines. For

the last experiment, our goal was to investigate if the "smoothness" between consecutive subspaces, enforced by the geodesic constraint, assists in dealing with measurement noise.

The results in this section used the parameters $n = 40$, $m = 20$, $r = 3$, $\ell = 1$, and $T = 80$, unless otherwise specified. For each 3-dimensional subspace, we observed only a single vector. To generate data from a geodesic, we first took two randomly generated orthonormal bases and applied Equation (16) to obtain $\mathbf{U}_i \in \mathcal{V}^{n \times r}$ for all $i \in [T]$. Then, in contrast to Equation (19), we generate measurements according to

$$\mathbf{y}_{i,j} = \mathbf{A}_{i,j}\mathbf{U}_i\mathbf{g}_{i,j} + \boldsymbol{\eta}_{i,j} \in \mathbb{R}^m, \qquad \forall j \in [\ell], \ \forall i \in [T], \tag{20}$$

where $\mathbf{g}_{i,j} \sim \mathcal{N}(0,\mathbf{I}_r)$, $\boldsymbol{\eta}_{i,j} \sim \mathcal{N}(0, \sigma_{\mathbf{y}} \cdot \mathbf{I}_m)$, and each column of $\mathbf{A}_{i,j} \in \mathbb{R}^{m \times n}$ is drawn iid from a Gaussian distribution (i.e., $\mathbf{a}_{i,j,k} \sim \mathcal{N}(0, \mathbf{I}_n)$) in the matrix sensing setting and sub-sampled diagonal matrices for the matrix completion setting. Here, $\sigma_{\mathbf{y}}$ is assumed to be 0 unless otherwise stated.

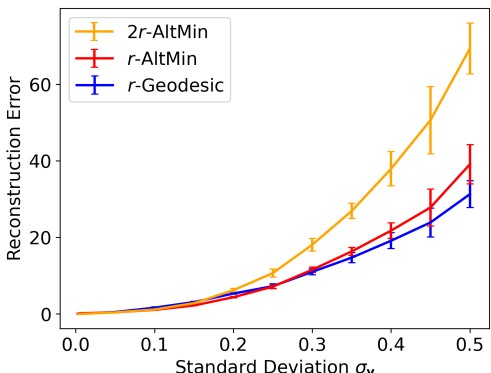

Figure 6 demonstrates that our algorithm can recover the ground truth data and subspace more quickly (in terms of iterations) using our spectral initialization method compared to random initialization. While random initialization often converges to poor local minima, spectral initialization often converges to a solution with near-zero subspace error. Nevertheless, this illustrates that even within noisy settings, our algorithm can faithfully estimate the data. Appendix B.1 illustrates the benefits of the spectral method and its failure cases.

Figures 5b and 5c show phase transitions similar to Figure 5a to obtain convergence to the ground truth solution. In Figure 5b, we vary $T$ and $r$ while fixing $m = 0.5n = 20$ and $\ell = 1$. Figure 5c is a similar setting, but with $\ell = r - 1$. Unlike the fully-sampled case, the phase transition is less clear in both cases and seem to require more samples once the rank $r$ gets closer to $d$. For the under-sampled cases, it seems that the number of data points at each time point $\ell$ plays an important

Figure 7: Reconstruction error (NRMSE) over $\sigma_{\mathbf{y}}$. For a larger noise variance, $2r$-AltMin seems to overfit the noise rapidly, whereas our method is more robust.

role in ensuring accurate recovery. Nevertheless, for our applications of interest, since we generally assume that $r \ll d$, we can still faithfully use our method with a smaller number of time points $T$. We leave a rigorous analysis of the relationship amongst the parameters for convergence for future work.

Lastly, we compare the performance of our algorithm compared to AltMin with a rank choice of $r$ and $2r$ with varying $\sigma_{\mathbf{y}}$ for matrix sensing. By enforcing the smoothness constraint via the geodesic, we expect our model to handle measurement noise more efficiently. Figure 7 verifies this conjecture: our algorithm obtains a much lower reconstruction error than $2r$-AltMin as a function of noise.

**fMRI Data Experiments.** Here, we present experiments on reconstructing dynamic fMRI images. We considered an oscillating steady state imaging (OSSI) dataset (Guo et al., 2020) that was acquired on a 3T GE MR750 scanner with a 32-channel head coil. The data were comprised of 167 slow-time, 10 fast-time, and $(128 \times 128)$ spatial samples. We considered a single point in fast time and 160 slow-time points, which gave us a dataset of 160 frames, where each frame was of dimension $(128 \times 128)$. We considered the slow-time points because we hypothesize that the slowly changing movements were suitable for a geodesic model.

To this end, we used an approximately 50% missing $k$-space sampling mask on each frame, where the center of the mask was fully observed but rows above and below the center were missing (see Figure 18 for examples). For our algorithm, we chose parameters $\ell = 4$ to match the slow time frames, which leads to $T = 40$ and $\lambda = 0.1$. We evenly sampled $t_i$ from the range $[0, 0.5]$ and chose the rank to be $r = 5$. For the baselines, we considered zero-filled reconstructions, data sharing, and $2r$-AltMin. For data sharing, we identified the nearest data sample that contained the observed row and used it to impute the missing row for each frame. We do not include results for $r$-AltMin, as it was generally superseded by $2r$-AltMin. For both AltMin and our algorithm, we used the zero-filled method as the initial starting point for our algorithm. This is to investigate if we can improve upon the nearest-neighbors method by enforcing the low-rank structure. Both

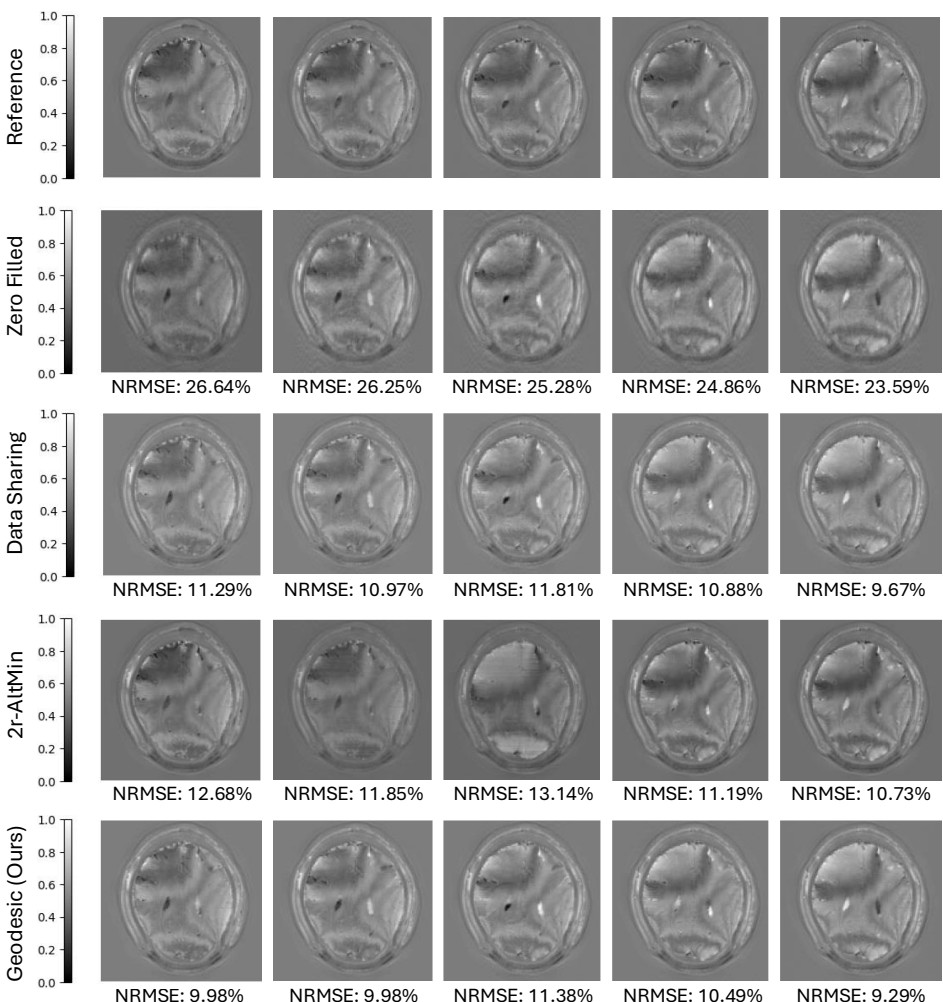

Figure 8: fMRI reconstruction results, where each frame was masked by a 50% sampled $k$-space mask. Both our algorithm and AltMin used a subspace dimension of $r = 5$, with the data-sharing method used for initialization. Figure 20 presents difference images between the methods and the reference images to more clearly illustrate each algorithm's performance.

algorithms were run until the change in the NRMSE between two successive iterations were less than $10^{-6}$, unless otherwise stated.

Figure 8 shows that the geodesic model significantly improved upon the nearest-neighbors method and achieved the lowest NRMSE scores. On the other hand, while $2r$-AltMin improved upon the initial point, there exist some time frames in which it performed worse. Appendix B.2 provides additional fMRI experiments, particularly on dynamic Shepp-Logan phantoms. Here, we expect AltMin to perform best since the Shepp-Logan phantoms have an *exact* (planted) low-rank structure. Nonetheless, our method performed closely to AltMin, highlighting the ability of our approach to reconstruct fMRI data.

**Video Data Experiments.** We considered two video datasets: a video of a mouse moving towards a camera and the plane video from Section 5.1. The mouse video has a total of 90 frames, each with dimensions $(70 \times 100)$. This video was also used in earlier works (Nayer et al., 2020; Kwon et al., 2022). Our goal was to see if we can improve upon the globally low-rank methods by incorporating a temporal component. Since these videos are neither exactly low-rank nor lie on a geodesic, we do not expect perfect recovery.

| Dataset | Method | $r = 3$ | | $r = 5$ | |
|---------|--------|---------|---|---------|---|
| | | PSNR ↑ | RMSE ↓ | PSNR ↑ | RMSE ↓ |
| Mouse | $r$-AltMin (Jain et al., 2013) | $20.55 \pm 0.030$ | $0.108 \pm 0.001$ | $22.05 \pm 0.061$ | $0.089 \pm 0.001$ |
| | $2r$-AltMin (Jain et al., 2013) | $\underline{22.23} \pm 0.080$ | $\underline{0.086} \pm 0.001$ | $\underline{22.92} \pm 0.098$ | $\mathbf{0.080} \pm 0.001$ |
| | $r$-GNMR (Zilber & Nadler, 2022) | $19.61 \pm 0.095$ | $0.128 \pm 0.003$ | $20.73 \pm 0.348$ | $0.114 \pm 0.011$ |
| | $2r$-GNMR (Zilber & Nadler, 2022) | $20.88 \pm 0.095$ | $0.115 \pm 0.014$ | $19.22 \pm 0.314$ | $0.129 \pm 0.010$ |
| | $r$-GROUSE (Balzano et al., 2010) | $19.59 \pm 0.143$ | $0.130 \pm 0.007$ | $20.43 \pm 0.431$ | $0.113 \pm 0.005$ |
| | $2r$-GROUSE (Balzano et al., 2010) | $20.86 \pm 0.168$ | $0.107 \pm 0.002$ | $19.97 \pm 0.436$ | $0.119 \pm 0.006$ |
| | $r$-Geodesic (Ours) | $\mathbf{22.70} \pm 0.158$ | $\mathbf{0.083} \pm 0.001$ | $\mathbf{23.28} \pm 0.164$ | $\underline{0.082} \pm 0.001$ |
| Plane | $r$-AltMin (Jain et al., 2013) | $25.27 \pm 0.058$ | $0.058 \pm 0.001$ | $25.25 \pm 0.047$ | $0.058 \pm 0.001$ |
| | $2r$-AltMin (Jain et al., 2013) | $\underline{28.47} \pm 0.274$ | $\mathbf{0.040} \pm 0.001$ | $28.61 \pm 0.114$ | $\underline{0.040} \pm 0.001$ |
| | $r$-GNMR (Zilber & Nadler, 2022) | $24.37 \pm 0.854$ | $0.072 \pm 0.022$ | $24.39 \pm 0.333$ | $0.067 \pm 0.002$ |
| | $2r$-GNMR (Zilber & Nadler, 2022) | $27.00 \pm 0.827$ | $0.061 \pm 0.014$ | $27.14 \pm 0.832$ | $0.055 \pm 0.006$ |
| | $r$-GROUSE (Balzano et al., 2010) | $24.31 \pm 0.297$ | $0.068 \pm 0.006$ | $26.62 \pm 0.324$ | $0.054 \pm 0.004$ |
| | $2r$-GROUSE (Balzano et al., 2010) | $27.46 \pm 0.693$ | $0.064 \pm 0.017$ | $\mathbf{30.17} \pm 0.396$ | $0.046 \pm 0.004$ |
| | $r$-Geodesic (Ours) | $\mathbf{29.01} \pm 0.207$ | $\underline{0.040} \pm 0.002$ | $\underline{29.20} \pm 0.379$ | $\mathbf{0.039} \pm 0.003$ |

Table 1: Quantitative results for varying ranks for different metrics for both datasets across 10 trials. Both datasets are corrupted with Gaussian noise of standard deviation $\sigma_{\mathbf{y}} = 0.01$, where for the mouse data we observe 50% of each frame, and for the plane data we observe 40% of each frame. Best results are bolded and second best results are underlined.

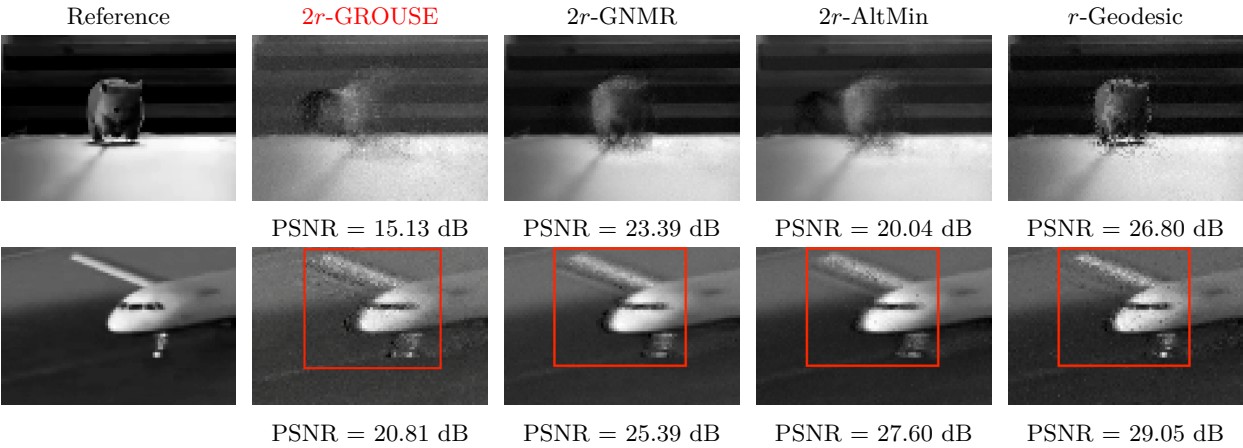

Figure 9: Visual results with corresponding PSNRs for a single frame of the mouse and plane video dataset with Gaussian noise with standard deviation $\sigma_{\mathbf{y}} = 0.01$. Top: Mouse frame with 50% missing pixels at random for $r = 5$. Bottom: Plane frame with 60% missing pixels at random for $r = 3$.

For the mouse video dataset, we consider the noisy matrix completion setting, in which approximately 50% of the pixels in each frame are missing and corrupted with Gaussian noise of $\sigma_{\mathbf{y}} = 0.01$. For the plane video dataset, we observe only 40% of the data, also with $\sigma_{\mathbf{y}} = 0.01$. Alongside AltMin, we consider the baseline GNMR (Zilber & Nadler, 2022), a matrix recovery algorithm based on Gauss-Newton linearization and GROUSE (Balzano et al., 2010), a Grassmannian rank-one subspace estimation algorithm. For both GNMR and GROUSE, we also vectorize and stack all the frames to construct a single matrix. For GROUSE, we ran a total of 50 iterations and selected the learning rate via a grid search over constant and diminishing step sizes. We picked the ranks to be $r = 3, 5$, where for our algorithm we set $\ell = r - 1$. This was to emphasize the fact that our algorithm can handle the case in which $\ell < r$. We evenly sample the timesteps $t_i \in [0.3, 0.7]$ and set $\lambda = 1.0$.

Figure 9 displays frames from both video data experiments. Intriguingly, we observe that our algorithm is less prone to the "blurring" effects commonly seen in models that use low-rank approximations. For example, in the bottom row of Figure 9, the plane appears less blurry with our algorithm, especially in the

areas highlighted by the red box. We see similar results in the mouse frame, where the mouse is clearly more prominent using our algorithm. In Table 1, we present quantitative results across all frames for our algorithm and the baselines. Overall, our algorithm outperforms the others in terms of PSNR and RMSE across different ranks. However, in some trials, AltMin and GROUSE slightly outperform our algorithm. We believe this is due to slightly more noise or "speckles" in the frames reconstructed by our method. This highlights a tradeoff: when the dataset does not lie on a geodesic, our algorithm more accurately captures the object of interest but may introduce additional noise, since it estimates more parameters than a static model. Moreover, GROUSE can model a time-varying $2r$-dimensional subspace, whereas the other baselines, including ours, learn a model within at most a fixed $2r$-dimensional subspace. This added flexibility explains why GROUSE achieves a higher PSNR in one of the instances.

### 5.3 Ablation Studies

This section presents ablation studies for our algorithm, focusing in particular on the choice of hyperparameters. To this end, we use the mouse video dataset and adopt the same parameters as in the undersampled case unless otherwise specified: $r = 5$, $\ell = 4$, timesteps $t_i \in [0.3, 0.7]$, $\lambda = 1.0$, and approximately 50% missing pixels when applicable.

**Effect of Hyperparameters.** First, we present the effect of the hyperparameters: the timesteps $t_i$ and the regularization parameter $\lambda$. For the timesteps, the most natural choice is to choose evenly spaced values from $t_i \in [0.0, 1.0]$, so that $\mathbf{U}_0$ corresponds to the initial subspace generated from $\mathbf{X}_0$ and $\mathbf{U}_T$ corresponds to the subspace generated from $\mathbf{X}_T$. However, in the undersampled setting, we observed that this choice can lead to lower PSNR at the two endpoints. We hypothesize that this occurs because the endpoint subspaces are estimated from undersampled observations and may therefore be less accurate; as a result, enforcing the geodesic constraint at or near the endpoints can amplify endpoint estimation errors. Figure 11 (left) demonstrates this effect and shows that choosing interior points $t_i \in [0.3, 0.7]$ improves performance. Thus, the timesteps are treated as hyperparameters. In practice, one can start with evenly spaced values $t_i \in [0.0, 1.0]$ and use cross-validation to choose the best values.

In Figure 11 (right), we present results demonstrating the effect of the parameter $\lambda$. When $\lambda$ is small, the geodesic constraint is weak, and the average PSNR across all frames is substantially lower. As $\lambda$ increases toward 1.0, performance improves, suggesting that the geodesic constraint provides useful regularization for recovering the underlying dynamic subspace structure.

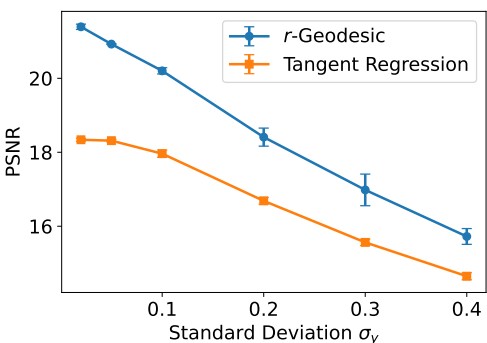

Figure 10: Comparison between the proposed $r$-Geodesic method and the tangent regression baseline under varying noise levels $\sigma_{\mathbf{y}}$.

**Case Study on $\ell > r$.** Throughout our experiments, we primarily choose $r > \ell$ to emphasize that our algorithm can effectively estimate the subspaces even when the target rank is larger than the number of observed data points at each time point. Next, we provide experimental results in the fully sampled setting that support the use of our algorithm even when $\ell > r$. In this setting, it is possible to estimate local subspaces $\widehat{\mathbf{U}}_i \in \mathbb{R}^{n \times r}$ independently at each time point by computing the top-$r$ left singular vectors of the observation matrix $\mathbf{Y}_i$. This enables another geodesic baseline: (i) first estimate the local subspaces, (ii) map them to the tangent space at a reference subspace $\mathbf{U}_{\text{ref}}$ using the logarithm map, (iii) fit an affine trajectory $\widehat{\mathbf{A}} + t_i \widehat{\mathbf{B}}$ in this tangent space, and (iv) map the fitted tangent vectors back to the Grassmannian using the exponential map. Here, $\widehat{\mathbf{A}}$ and $\widehat{\mathbf{B}}$ are the fitted intercept and tangent direction, respectively; $\widehat{\mathbf{A}}$ determines the offset of the trajectory in the tangent space, while $\widehat{\mathbf{B}}$ controls how the subspace evolves over time. Since this curve is obtained by mapping this affine tangent-space trajectory back to the Grassmannian, this baseline allows the subspace to vary over time while constraining its evolution to be approximately geodesic in the tangent coordinates. We defer implementation details to Appendix B.2.

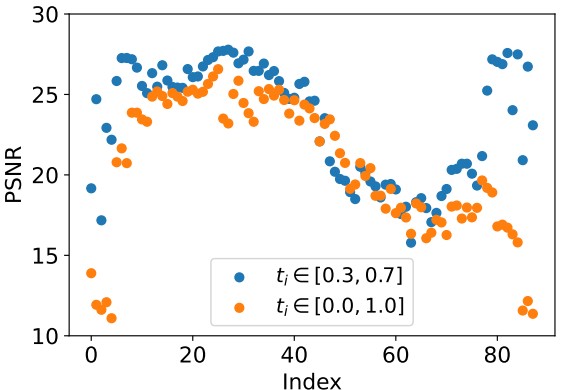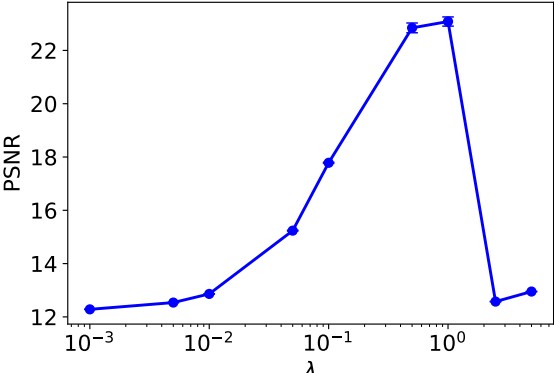

Figure 11: Effect of hyperparameters: timesteps $t_i$ and regularization parameter $\lambda$. Left: We compare choosing timesteps $t_i \in [0.3, 0.7]$ with $t_i \in [0.0, 1.0]$. Using the full interval leads to lower PSNR near the endpoint frames, while restricting the timesteps to the interior mitigates this degradation. Right: We study the effect of the regularization parameter $\lambda$. Smaller values of $\lambda$, corresponding to a weaker geodesic constraint, yield substantially lower average PSNR across frames. Increasing $\lambda$ toward 1.0 improves performance, supporting the benefit of the geodesic constraint.

Figure 10 shows that our method outperforms this baseline for varying values of the noise standard deviation $\sigma_{\mathbf{y}}$, using timesteps $t_i \in [0.3, 0.7]$. For the baseline, we construct the reference subspace by pooling all of the data and computing the top-$r$ left singular vectors, which corresponds to the static method. We hypothesize that the improvement comes from the fact that the baseline separates local subspace estimation from the geodesic fitting step. As a result, errors in the independently estimated local subspaces can propagate directly into the fitted trajectory, especially when the per-time-point observations are noisy or limited. In contrast, our method jointly estimates the dynamic subspace trajectory by incorporating the geodesic constraint directly into the reconstruction objective. This approach allows observations across time to inform one another through the shared geometric structure, leading to a more stable estimate of the evolving subspaces. Thus, our method outperforms this tangent-space geodesic baseline even in the favorable regime $\ell > r$, where local subspace estimation is possible.

## 6 CONCLUSION AND FUTURE WORK

This work proposed an RBMM algorithm for estimating low-dimensional, time-varying data from undersampled measurements. Our algorithm constrains the time-varying subspaces to follow a geodesic on the Grassmannian. We demonstrated that our algorithm can effectively handle measurement noise and missing data compared to existing baselines in both synthetic and real settings. We also showed that our algorithm enjoys two convergence properties: a monotonically decreasing objective function and convergence to a stationary point in $\widetilde{\mathcal{O}}(\epsilon^{-2})$, where $\epsilon > 0$ is an error term representing the norm of the gradients for each block.

There are several exciting avenues for future work. While our experiments demonstrated that our model and algorithm are already useful and applicable to real data, it would be of interest to develop a continuous piecewise-geodesic model rather than the single geodesic considered here. This extension may improve applicability across a wider range of data types, and our work provides a key building block toward such a piecewise model. Furthermore, for fMRI data, the measurements are intrinsically tensor-valued, whereas our current approach vectorizes each image for reconstruction. Extending our algorithm to the tensor case could better exploit spatial structure for faster and more accurate reconstruction.

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

## A    Derivation of Block Update Steps

This section provides detailed derivations of the block update steps for Algorithm 1. Recall that

$$\mathbf{U}_i = \mathbf{H}\cos(\mathbf{\Theta}t_i) + \mathbf{Z}\sin(\mathbf{\Theta}t_i) = \mathbf{Q}\mathbf{R}_i, \quad \text{where } \mathbf{Q} \coloneqq [\mathbf{H}, \mathbf{Z}], \ \mathbf{R}_i \coloneqq \begin{bmatrix} \cos(\mathbf{\Theta}t_i) \\ \sin(\mathbf{\Theta}t_i) \end{bmatrix}.$$

### A.1    Updates for X

Suppose $\mathbf{U}_i = \mathbf{Q}\mathbf{R}_i$ are fixed and let us define

$$f(\mathbf{x}_{i,j}) = \frac{1}{2} \left\| \widetilde{\mathbf{y}}_{i,j} - \widetilde{\mathbf{A}}_{i,j}\mathbf{x}_{i,j} \right\|_2^2. \tag{21}$$

We construct a quadratic majorizer for $f$ at each iteration $k$, which we will denote as $g^k$ for $\mathbf{x}_{i,j}^k$. Following the second-order Taylor expansion of $f$ at $\mathbf{x}_{i,j}^k$, we have

$$f(\mathbf{x}_{i,j}) = f(\mathbf{x}_{i,j}^k) + \nabla f(\mathbf{x}_{i,j}^k)^\top (\mathbf{x}_{i,j} - \mathbf{x}_{i,j}^k) + \frac{L}{2} \|\mathbf{x}_{i,j} - \mathbf{x}_{i,j}^k\|_2^2 \tag{22}$$

$$\leq f(\mathbf{x}_{i,j}^k) + \nabla f(\mathbf{x}_{i,j}^k)^\top (\mathbf{x}_{i,j} - \mathbf{x}_{i,j}^k) + \frac{L}{2} \|\mathbf{x}_{i,j} - \mathbf{x}_{i,j}^k\|_2^2 \underbrace{+ \frac{\lambda_x}{2} \|\mathbf{x}_{i,j} - \mathbf{x}_{i,j}^k\|_2^2}_{\text{to ensure quadratic gap}} \tag{23}$$

$$= f(\mathbf{x}_{i,j}^k) + \nabla f(\mathbf{x}_{i,j}^k)^\top (\mathbf{x}_{i,j} - \mathbf{x}_{i,j}^k) + \frac{L + \lambda_x}{2} \|\mathbf{x}_{i,j} - \mathbf{x}_{i,j}^k\|_2^2 \tag{24}$$

$$= \underbrace{\frac{L + \lambda_x}{2} \left\| \mathbf{x}_{i,j} - \left( \mathbf{x}_{i,j}^k - \frac{1}{L + \lambda_x} \nabla f(\mathbf{x}_{i,j}^k) \right) \right\|_2^2}_{=:g^k(\mathbf{x}_{i,j})} + \underbrace{\left( f(\mathbf{x}_{i,j}^k) - \frac{1}{2(L + \lambda_x)} \|\nabla f(\mathbf{x}_{i,j}^k)\|_2^2 \right)}_{=:c}, \tag{25}$$

where $c$ is a constant independent of $\mathbf{x}_{i,j}$. Thus, we have

$$\mathbf{x}_{i,j}^{k+1} = \arg\min_{\mathbf{x}_{i,j}} g^k(\mathbf{x}_{i,j}; \mathbf{x}_{i,j}^k) = \arg\min_{\mathbf{x}_{i,j}} \left\| \mathbf{x}_{i,j} - \left( \mathbf{x}_{i,j}^k - \frac{1}{L + \lambda_x} \nabla f(\mathbf{x}_{i,j}^k) \right) \right\|_2^2$$

$$= \mathbf{x}_{i,j}^k - \frac{1}{L + \lambda_x} \nabla f(\mathbf{x}_{i,j}^k).$$

### A.2    Updates for Q

To update $\mathbf{Q}$, notice that the data consistency term in Equation (4) is not a function of $\mathbf{Q}$, so we only need to consider the subspace regularizer. Define

$$\hat{\mathbf{Q}} = \arg\min_{\mathbf{Q}} f(\mathbf{Q}) = \arg\min_{\mathbf{Q}} \frac{\lambda}{2} \sum_{i=1}^{T} \sum_{j=1}^{\ell} \left\| \left( \mathbf{Q}\mathbf{R}_i\mathbf{R}_i^\top\mathbf{Q}^\top - \mathbf{I}_d \right) \mathbf{x}_{i,j} \right\|_2^2 \tag{26}$$

$$= \arg\min_{\mathbf{Q}} \frac{\lambda}{2} \sum_{i=1}^{T} \left\| \left( \mathbf{Q}\mathbf{R}_i\mathbf{R}_i^\top\mathbf{Q}^\top - \mathbf{I}_d \right) \mathbf{X}_i \right\|_{\mathsf{F}}^2. \tag{27}$$

We can further simplify the objective function in Equation (7) into

$$\arg\min_{\mathbf{Q}} f(\mathbf{Q}) = \arg\min_{\mathbf{Q}} \frac{\lambda}{2} \sum_{i=1}^{T} \left\| \left( \mathbf{Q}\mathbf{R}_i\mathbf{R}_i^\top\mathbf{Q}^\top - \mathbf{I}_d \right) \mathbf{X}_i \right\|_{\mathsf{F}}^2$$

$$= \arg\min_{\mathbf{Q}} \frac{\lambda}{2} \sum_{i=1}^{T} - \operatorname{tr} \left( \mathbf{X}_i^\top \mathbf{Q}\mathbf{R}_i\mathbf{R}_i^\top\mathbf{Q}^\top\mathbf{X}_i \right) + \operatorname{tr} \left( \mathbf{X}^\top\mathbf{X} \right)$$

$$= \arg\min_{\mathbf{Q}} - \frac{\lambda}{2} \sum_{i=1}^{T} \left\| \mathbf{X}_i^\top \mathbf{Q}\mathbf{R}_i \right\|_{\mathsf{F}}^2,$$

as $\text{tr}\left(\mathbf{X}^\top\mathbf{X}\right)$ is a constant. Then, the gradient of $f$ with respect to $\mathbf{Q}$ is given by

$$\nabla f_{\mathbf{Q}}(\mathbf{Q}) = -\lambda \sum_{i=1}^{T} \mathbf{X}_i \mathbf{X}_i^\top \mathbf{Q} \mathbf{R}_i \mathbf{R}_i^\top. \tag{28}$$

For the $k$-th iteration, let us fix $\mathbf{x}_{i,j}^k$ and $\mathbf{\Theta}_i^k$ and consider the following majorizer

$$g^k(\mathbf{Q}) := \frac{1}{2}\text{tr}\left(\mathbf{Q}^\top \nabla f_{\mathbf{Q}}(\mathbf{Q})\right) \underbrace{+\frac{\lambda_{\mathbf{Q}}}{4}\|\mathbf{Q}-\mathbf{Q}^k\|_2^2}_{\text{to ensure quadratic gap}}. \tag{29}$$

Notice that

$$f(\mathbf{Q}^k) = g^k(\mathbf{Q}^k) \quad \text{and} \quad f(\mathbf{Q}) \le g^k(\mathbf{Q}) \quad \forall \mathbf{Q} \in \mathbb{R}^{n\times 2r},$$

by the first-order Taylor approximation, and hence $g^k(\mathbf{Q})$ is a valid majorizer. We can simplify the majorizer into

$$\begin{aligned} g^k(\mathbf{Q}) &= \frac{1}{2}\text{tr}\left(\mathbf{Q}^\top \nabla f_{\mathbf{Q}}(\mathbf{Q})\right) - \frac{\lambda_{\mathbf{Q}}}{2}\text{tr}\left(\mathbf{Q}^\top\mathbf{Q}^k\right) \\ &= \frac{1}{2}\text{tr}\left(\mathbf{Q}^\top\left(\nabla f_{\mathbf{Q}}(\mathbf{Q}) - \lambda_{\mathbf{Q}}\mathbf{Q}^k\right)\right), \end{aligned}$$

where we have used the fact that $\mathbf{Q}^\top\mathbf{Q} = \mathbf{Q}^{k\top}\mathbf{Q}^k = \mathbf{I}_d$. Then, we have

$$\mathbf{Q}^{k+1} = \operatorname*{arg\,min}_{\mathbf{Q}\in\mathcal{V}^{n\times 2r}} g^k(\mathbf{Q}) = \left\|\mathbf{Q} - \left(\lambda_{\mathbf{Q}}\mathbf{Q}^k - \nabla f_{\mathbf{Q}}(\mathbf{Q})\right)\right\|_{\mathsf{F}}^2 \tag{30}$$

$$= \mathbf{W}\mathbf{V}^\top, \quad \text{where } \left(\lambda_{\mathbf{Q}}\mathbf{Q}^k - \nabla f_{\mathbf{Q}}(\mathbf{Q})\right) = \mathbf{W}\mathbf{\Sigma}\mathbf{V}^\top, \tag{31}$$

where the last equality comes from the fact that we are retracting onto the Stiefel manifold (Absil et al., 2007; Higham, 1989).

### A.3   Updates for $\mathbf{\Theta}$

Similarly, let us consider the simplified objective function

$$\hat{\mathbf{\Theta}} = \operatorname*{arg\,min}_{\mathbf{\Theta}} f(\mathbf{\Theta}) = \operatorname*{arg\,min}_{\mathbf{\Theta}} -\frac{\lambda}{2}\sum_{i=1}^{T}\left\|\mathbf{X}_i^\top\mathbf{Q}\mathbf{R}_i\right\|_{\mathsf{F}}^2. \tag{32}$$

Recall that $\boldsymbol{\Theta} \in \mathbb{R}^{r \times r}$ is a diagonal matrix. To preserve the structure of $\boldsymbol{\Theta}$, we update in sequence each coordinate of $\boldsymbol{\Theta}$, which we denote as $\theta_d \in \mathbb{R}$, $\forall d \in [r]$. We first simplify the loss function as follows:

$$
\begin{aligned}
\hat{\boldsymbol{\Theta}} &= \arg\min_{\boldsymbol{\Theta}} \; -\frac{\lambda}{2} \sum_{i=1}^{T} \left\| \mathbf{X}_i^\top \mathbf{Q} \mathbf{R}_i \right\|_{\mathsf{F}}^2 = -\frac{\lambda}{2} \sum_{i=1}^{T} \mathrm{tr}\left\{ \mathbf{X}_i^\top \mathbf{Q} \mathbf{R}_i \mathbf{R}_i^\top \mathbf{Q}^\top \mathbf{X}_i \right\} \\
&= \arg\min_{\boldsymbol{\Theta}} \; -\frac{\lambda}{2} \sum_{i=1}^{T} \mathrm{tr}\left\{ \mathbf{X}_i^\top \left( \mathbf{H} \cos^2(\boldsymbol{\Theta} t_i) \mathbf{H}^\top + 2\{ \mathbf{H} \cos(\boldsymbol{\Theta} t_i) \sin(\boldsymbol{\Theta} t_i) \mathbf{Z}^\top \} \right. \right. \\
&\hspace{8cm} \left. \left. + \; \mathbf{Z} \sin^2(\boldsymbol{\Theta} t_i) \mathbf{Z}^\top \right) \mathbf{X}_i \right\} \\
&= \arg\min_{\boldsymbol{\Theta}} \; -\frac{\lambda}{2} \sum_{i=1}^{T} \mathrm{tr}\left\{ \cos^2(\boldsymbol{\Theta} t_i) \cdot \left( \mathbf{H}^\top \mathbf{X}_i \mathbf{X}_i^\top \mathbf{H} \right) + 2\cos(\boldsymbol{\Theta} t_i) \sin(\boldsymbol{\Theta} t_i) \cdot \{ \mathbf{Z}^\top \mathbf{X}_i \mathbf{X}_i^\top \mathbf{H} \} \right. \\
&\hspace{8cm} \left. + \sin^2(\boldsymbol{\Theta} t_i) \cdot \left( \mathbf{Z}^\top \mathbf{X}_i \mathbf{X}_i^\top \mathbf{Z} \right) \right\} \\
&= \arg\min_{\theta_d} \; -\frac{\lambda}{2} \sum_{i=1}^{T} \sum_{d=1}^{r} \mathrm{tr}\left\{ \cos^2(\theta_d t_i) \cdot \left[ \mathbf{H}^\top \mathbf{X}_i \mathbf{X}_i^\top \mathbf{H} \right]_{d,d} \right. \\
&\hspace{4cm} + 2\cos(\theta_d t_i) \sin(\theta_d t_i) \cdot \left[ \{ \mathbf{Z}^\top \mathbf{X}_i \mathbf{X}_i^\top \mathbf{H} \} \right]_{d,d} \\
&\hspace{6cm} \left. + \sin^2(\theta_d t_i) \cdot \left[ \mathbf{Z}^\top \mathbf{X}_i \mathbf{X}_i^\top \mathbf{Z} \right]_{d,d} \right\} \\
&= \arg\min_{\theta_d} \; -\frac{\lambda}{2} \sum_{i=1}^{T} \sum_{d=1}^{r} \alpha_{i,s} \cos^2(\theta_d t_i) + 2\beta_{i,s} \cos(\theta_d t_i) \sin(\theta_d t_i) + \gamma_{i,s} \sin^2(\theta_d t_i),
\end{aligned}
$$

where we define

$$
\begin{aligned}
\alpha_{i,s} &:= \left[ \mathbf{H}^\top \mathbf{X}_i \mathbf{X}_i^\top \mathbf{H} \right]_{d,d} \\
\beta_{i,s} &:= \mathrm{real}\left\{ \left[ \mathbf{Z}^\top \mathbf{X}_i \mathbf{X}_i^\top \mathbf{H} \right]_{d,d} \right\} \\
\gamma_{i,s} &:= \left[ \mathbf{Z}^\top \mathbf{X}_i \mathbf{X}_i^\top \mathbf{Z} \right]_{d,d}.
\end{aligned}
$$

Now, for some constants $a, b, x \in \mathbb{R}$, consider the trigonometric identities

$$
\begin{aligned}
2\cos(x)\sin(x) &= \sin(2x) \\
\cos^2(x) + \sin^2(x) &= 1 \\
\cos^2(x) &= \frac{1}{2}\left( \cos(2x) + 1 \right) \\
a\cos(x) + b\sin(x) &= \sqrt{a^2 + b^2} \cos(x - \mathrm{arctan2}(b, a)).
\end{aligned}
$$

Using these properties, we can further simplify the loss:

$$
\begin{aligned}
\hat{\theta}_d &= \arg\min_{\theta_d} -\frac{\lambda}{2} \sum_{i=1}^{T} \sum_{d=1}^{r} \alpha_{i,s} \cos^2(\theta_d t_i) + 2\beta_{i,s} \cos(\theta_d t_i) \sin(\theta_d t_i) + \gamma_{i,s} \sin^2(\theta_d t_i) \\
&= \arg\min_{\theta_d} -\frac{\lambda}{2} \sum_{i=1}^{T} \sum_{d=1}^{r} \alpha_{i,s} \cos^2(\theta_d t_i) + \beta_{i,s} \sin(2\theta_d t_i) + \gamma_{i,s} \sin^2(\theta_d t_i) \\
&= \arg\min_{\theta_d} -\frac{\lambda}{2} \sum_{i=1}^{T} \sum_{d=1}^{r} \alpha_{i,s} \cos^2(\theta_d t_i) + \beta_{i,s} \sin(2\theta_d t_i) + \gamma_{i,s} \sin^2(\theta_d t_i) \\
&\qquad\qquad\qquad\qquad\qquad\qquad + \underbrace{\gamma_{i,s} \cos^2(\theta_d t_i) - \gamma_{i,s} \cos^2(\theta_d t_i)}_{\text{adding a zero}} \\
&= \arg\min_{\theta_d} -\frac{\lambda}{2} \sum_{i=1}^{T} \sum_{d=1}^{r} (\alpha_{i,s} - \gamma_{i,s}) \cos^2(\theta_d t_i) + \beta_{i,s} \sin(2\theta_d t_i) \\
&= \arg\min_{\theta_d} -\frac{\lambda}{2} \sum_{i=1}^{T} \sum_{d=1}^{r} \frac{\alpha_{i,s} - \gamma_{i,s}}{2} (\cos(2\theta_d t_i) + 1) + \beta_{i,s} \sin(2\theta_d t_i) \\
&= \arg\min_{\theta_d} -\frac{\lambda}{2} \sum_{i=1}^{T} \sum_{d=1}^{r} \frac{\alpha_{i,s} - \gamma_{i,s}}{2} \cos(2\theta_d t_i) + \beta_{i,s} \sin(2\theta_d t_i) \\
&= \arg\min_{\theta_d} -\frac{\lambda}{2} \sum_{i=1}^{T} \sum_{d=1}^{r} r_{i,s} \cdot \cos(2\theta_d t_i - \phi_{i,s}),
\end{aligned}
$$

where we define

$$
r_{i,s} := \sqrt{\left(\frac{\alpha_{i,s} - \gamma_{i,s}}{2}\right)^2 + \beta_{i,s}^2}
$$

$$
\phi_{i,s} := \operatorname{arctan2}\left(\beta_{i,s}, \frac{\alpha_{i,s} - \gamma_{i,s}}{2}\right)
$$

Thus, we have

$$
\hat{\theta}_d = \arg\min_{\theta_d} \sum_{i=1}^{T} f_{i,s}(\theta_d) = \arg\min_{\theta_d} -\frac{\lambda}{2} \sum_{i=1}^{T} r_{i,s} \cdot \cos(2\theta_d t_i - \phi_{i,s}).
$$

To construct a majorizer $g_{i,s}^k(\theta_d)$ for $f_{i,s}(\theta_d)$ at iteration $k$, we can consider a quadratic of the form

$$
g_{i,s}(\theta_d; \theta_d^k) := f_{i,s}(\theta_d^k) + \dot{f}_{i,s}(\theta_d^k) \cdot (\theta_d - \theta_d^k) + \frac{1}{2} w_{f_{i,s}}(\theta_d^k) \cdot (\theta_d - \theta_d^k)^2 + \underbrace{\frac{\lambda_\Theta}{2} (\theta_d - \theta_d^k)^2}_{\text{to ensure quadratic gap}},
$$

where $\dot{f}_{i,s}(\theta_d^k) = \lambda r_{i,s} t_i \sin(2\theta_d t_i - \phi_{i,s})$ is the derivative of $f_{i,s}$ and $w_{f_{i,s}}$ is an appropriate curvature (or weighting) function. Notice that a simple alternative for $w_{f_{i,s}}$ is to set $w_{f_{i,s}} = 2\lambda r_{i,s} t_i^2$, which is the Lipschitz constant of the derivative. However, we can construct a tighter majorizer by using the properties of our function. Notice that $\theta_d = \frac{\phi_{i,s}}{2t_i}$ is a minimizer of our function, and is quasi-convex on the interval $[\frac{\phi_{i,s} - \pi}{2t_i}, \frac{\phi_{i,s} + \pi}{2t_i}]$ about this point. Following Funai et al. (2008), we construct a curvature function $\bar{w}_{f_{i,s}}$ for points within this interval and periodically extend it to construct the final $w_{f_{i,s}}$. Since $f_{i,s}$ is symmetric about $\frac{\phi_{i,s}}{2t_i}$, or equivalently when

$$
g_{i,s}\left(\frac{\phi_{i,s}}{2t_i}; \theta_d^k\right) = 0.
$$

Then, solving for $\bar{w}_{f_{i,s}}(\theta_d^k)$ yields

$$\bar{w}_{f_{i,s}}(\theta_d^k) = \frac{\dot{f}_{i,s}(\theta_d^k)}{\theta_d^k - \frac{\phi_{i,s}}{2t_i}} + \lambda_{\boldsymbol{\Theta}},$$

which can be viewed as a translated Huber curvature function (with a small perturbation due to $\lambda_{\boldsymbol{\Theta}}$). For the case in which $\theta_d^k = \frac{\phi_{i,s}}{2t_i}$, we can let $\bar{w}_{f_{i,s}} = 2\lambda r_{i,s} t_i^2$. To form $w_{f_{i,s}}$, we need to make the denominator periodic, as the numerator is already periodic. The resulting periodic version of this curvature function is

$$w_{f_{i,s}}(\theta_d^k) = \begin{cases} \frac{\dot{f}_{i,s}(\theta_d^k)}{\text{mod}\left(\left(\theta_d^k - \frac{\phi_{i,s}}{2t_i}\right) + \frac{\pi}{2t_i}, \frac{2\pi}{2t_i}\right) - \frac{\pi}{2t_i}} + \lambda_{\boldsymbol{\Theta}} & \theta_d^k \neq \frac{\phi_{i,s} + 2\pi h}{2t_i}, h \in \mathbb{Z} \\ 2\lambda r_{i,s} t_i^2 + \lambda_{\boldsymbol{\Theta}} & \theta_d^k = \frac{\phi_{i,s} + 2\pi h}{2t_i}, h \in \mathbb{Z}. \end{cases}$$

The final majorizer that we obtain is

$$g_i(\theta_d; \theta_d^k) = \sum_{i=1}^{T} g_{i,s}(\theta_d; \theta_d^k).$$

Lastly, we can simply minimize this function by setting its derivative to zero and solving for $\theta_d$:

$$\left(\sum_{i=1}^{T} \dot{f}_{i,s}(\theta_d^k)\right) + \left(\sum_{i=1}^{T} w_{f_{i,s}}(\theta_d^k)\right) \cdot (\theta_d - \theta_d^k),$$

which yields the descent scheme

$$\theta_d^{k+1} = \theta_d^k - \sum_{i=1}^{T} \frac{\dot{f}_{i,s}(\theta_d^k)}{w_{f_{i,s}}(\theta_d^k)}. \tag{33}$$

One can interpret this descent scheme as a gradient descent step with a variable step size $w_{f_{i,s}}$.

We summarize the algorithm steps in Algorithm 1.

## B   Additional Experiments and Experimental Setup

This section provides additional experiments to supplement those in the main paper. Section B.1 provides more results on synthetic datasets, whereas Section B.2 provides details on the globally low-rank baseline methods, along with additional results on video and fMRI datasets. All experiments were done using a Macbook Pro with a Apple M2 Pro Chip.

### B.1   Synthetic Experiments

**Subspace Tracking Comparison.**   To demonstrate the effectiveness of our model, we compare our algorithm against several online subspace tracking algorithms in a planted geodesic setting under the fully sampled case. We choose $n = 20, r = 2, \ell = 1$ for two different noise levels $\sigma = 10^{-2}$ and $\sigma = 10^{-5}$ and compare to algorithms GROUSE (Balzano et al., 2010), Oja's algorithm (Oja, 1982), ISVD (Moonen et al., 1992), and PETRELS (Chi et al., 2012). In Figure 12, we show that our proposed method consistently shows better performance (lower error) on geodesic data.

**Effect of the Proximal Parameters.**   Firstly, we provide additional synthetic experiments, where we consider a planted geodesic setting under the same setup as in Section 5.2. Here, our objective to illustrate that the proximal parameters $\lambda_{\mathbf{Q}}, \lambda_{\boldsymbol{\Theta}}, \lambda_{\mathbf{x}}$ do not play a large role in the actual performance of the algorithm and are merely artifacts of our analyses. In Figure 13, we observe the changes in the subspace error and objective function value across all RBMM iterations for varying values of $\lambda_{\mathbf{Q}}, \lambda_{\boldsymbol{\Theta}}, \lambda_{\mathbf{x}}$. Clearly, we can see that the trajectory with respect to each parameter closely follow each other, as they are overlayed in Figure 13. For this experiment, we start with the same initialization for each parameter, such that the initialization does not bias the performance. Based on these results, we do not tune these parameters and simply set $\lambda_{\mathbf{Q}}, \lambda_{\boldsymbol{\Theta}}, \lambda_{\mathbf{x}} = 0$ across all experiments.

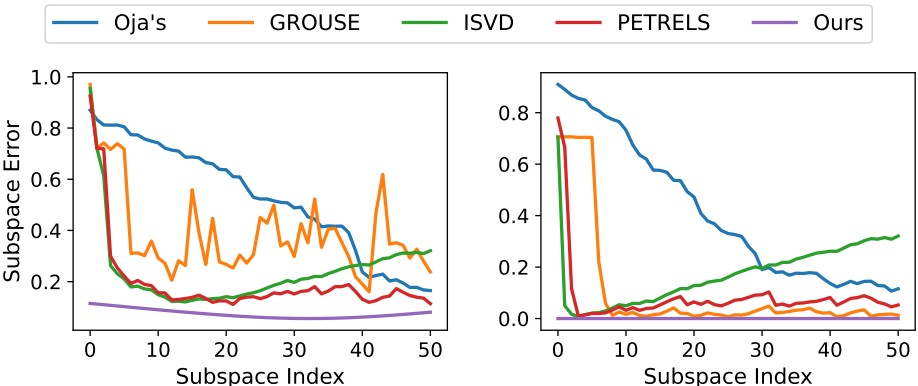

Figure 12: Subspace error at each time point for different subspace tracking algorithms for two different noise levels, $\sigma = 10^{-2}$ (left) and $\sigma = 10^{-5}$ (right). Note that the online algorithms estimate a single time point at a time, while the proposed method considers all time points.

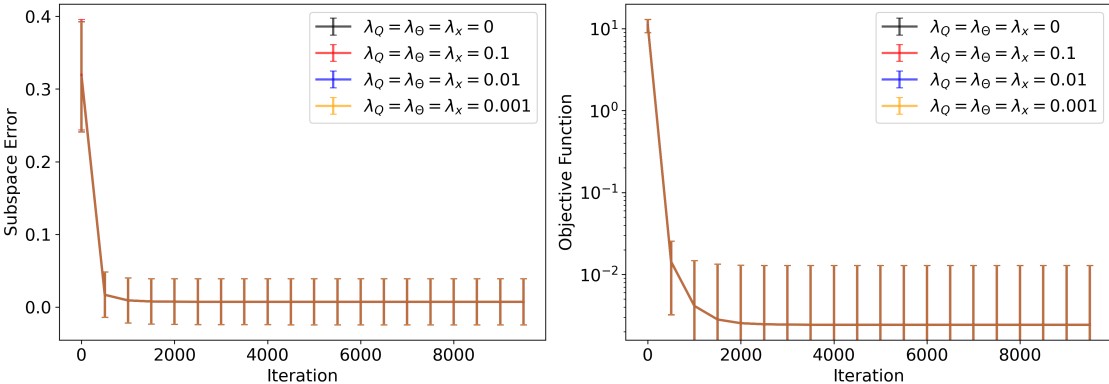

Figure 13: Experimental results observing the effect of varying the regularization (or proximal) parameters with $\lambda = 1$ and $\sigma_{\mathbf{y}} = 0.01$ averaged over 10 runs. Even for different values of $\lambda_{\mathbf{Q}}$, $\lambda_{\mathbf{\Theta}}$ and $\lambda_{\mathbf{x}}$, the subspace errors and objective function values closely overlap each other, demonstrating the robustness of these hyper-parameters.

**Visualizing Spectral Initialization.** Here, we aim to provide a visual representation of the benefit of using spectral initialization over random initialization in the matrix completion setting. To do so, we fix $\mathbf{H} \in \mathcal{V}^{d \times 3}$ and $\mathbf{Z} \in \mathcal{V}^{d \times 3}$ to be the top-6 principal components of FairFace dataset (Kärkkäinen & Joo, 2021) and generate $\mathbf{U}_i$ with $T = 20$ and $t_i \in [0, 1]$. Then, we generate synthetic data by computing

$$\mathbf{x}_i = \mathbf{U}_i \mathbf{g}_i, \quad \forall i \in [T],$$

where $\mathbf{g}_i \sim \mathcal{N}(0, \mathbf{I}_r)$. Then, we sub-sample 50% of the entries of each $\mathbf{x}_i$, by generating masking matrices $\mathbf{A}_i$ and computing

$$\mathbf{y}_i = \mathbf{A}_i \mathbf{U}_i \mathbf{g}_i, \quad \forall i \in [T].$$

Finally, we use the spectral initialization scheme outlined in Algorithm 1 to obtain $\mathbf{H}_0$ and $\mathbf{Z}_0$ and display their columns in Figure 14. In Figure 14, we can see that the subspaces obtained from the spectral initialization method resemble the true principal components, which suggests why spectral initialization is less likely to get stuck in local minima compared to random initialization.

**Convergence to Local Minima.** For this experiment, we use the same setting as in the previous section, where we consider a planted geodesic scenario with the true subspaces as the principal components of the

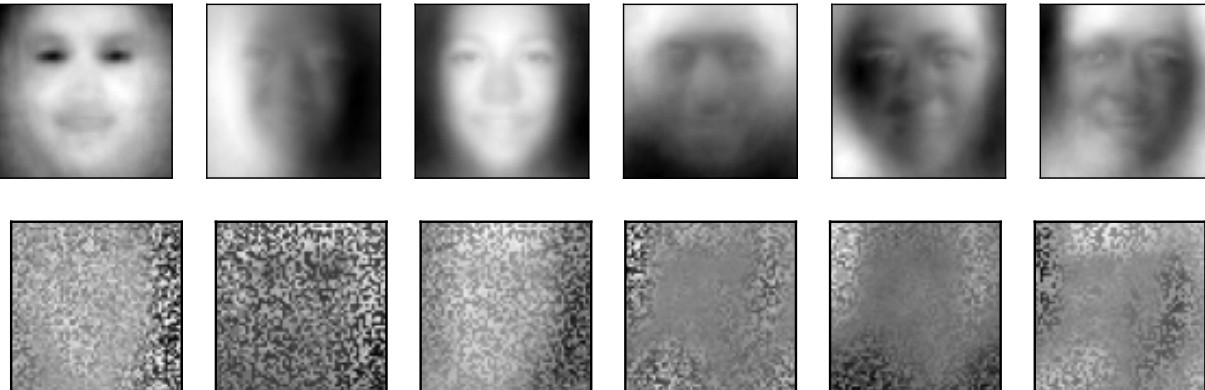

Figure 14: Visualization of the initial subspaces obtained by the spectral initialization technique outlined in Algorithm 1 with 50% missing data. Top: the true top-6 principal components of the FairFace dataset. Bottom: the geodesic subspaces obtained by the spectral initialization method. The initial subspaces seem to resemble actual faces, explaining why the method converges to the ground truth more often than random initialization.

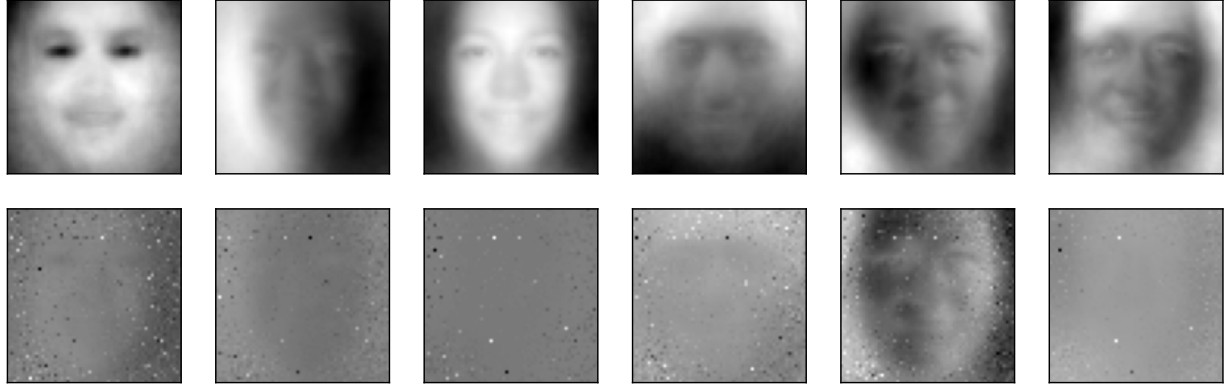

Figure 15: Visualization of the geodesic subspaces when the algorithm converges to a local minimum. Top: the true top-6 principal components of the FairFace dataset. Bottom: the geodesic subspaces at local convergence. Although a few subspaces resemble the true principal components, a few do not.

FairFace dataset. Our goal in this section is to visualize the resulting subspaces when the algorithm gets stuck in a local minimum (i.e., when the objective function value plateaus, but the subspace recovery error increases) to help better understand why the algorithm converges to such points. We take the subspaces (i.e., the resulting $\mathbf{H}$ and $\mathbf{Z}$) in one of these instances, and plot them in Figure 15. In Figure 15, we observe that many of the principal components do resemble the true ones but appear to be mostly submerged in noise. Interestingly, it appears that the 5-th component of the geodesic subspace has converged to the 6-th true component. There may have been some mixing, which could have caused the algorithm to converge to a sub-optimal solution. We leave a closer investigation of this phenomenon for future work.

## B.2 Experimental Setup and Real Experiments

Next, we present additional experiments on another video dataset and include more frames from the mouse video experiment. Additionally, we provide details on the globally low-rank baselines.

**Baselines.** Let $\Omega$ denote a binary masking matrix, where 1 denotes an observed entry and 0 denotes a missing entry. By concatenating all of the samples across time points, we can express our observed matrix (or sub-sampled matrix) as

$$\mathbf{Y} = \Omega \odot \mathbf{X}, \quad \text{where} \quad \mathbf{X} = [\mathbf{X}_1 \, \mathbf{X}_2 \dots \mathbf{X}_T] \in \mathbb{R}^{n \times \ell T},$$

where $\odot$ denotes the Hadamard product. For AltMin, we model $\mathbf{X}$ as a low-rank matrix by solving

$$\widehat{\mathbf{U}}, \widehat{\mathbf{V}} \in \underset{\mathbf{U}, \mathbf{V}}{\arg\min} \frac{1}{2} \|\mathbf{Y} - \Omega \odot (\mathbf{U}\mathbf{V}^\top)\|_\mathsf{F}^2.$$

We use alternating minimization to solve for $\mathbf{U}$ and $\mathbf{V}$, where each minimization problem involves using gradient descent with learning rate $\eta$, and was terminated once the difference between consecutive NRMSE values reached below $10^{-6}$. For the video data experiments, we used a spectral initialization scheme similar to our algorithm. Recall that we can construct a surrogate matrix $\hat{\mathbf{X}}$, which serves as an estimate for $\mathbf{X}$. We take the SVD of $\mathbf{X} = \widetilde{\mathbf{U}}\mathbf{\Lambda}\widetilde{\mathbf{V}}^\top$ and use

$$\mathbf{U}_0 = \widetilde{\mathbf{U}}_r \mathbf{\Lambda}_r^{1/2} \quad \text{and} \quad \mathbf{V}_0 = \widetilde{\mathbf{V}}_r \mathbf{\Lambda}_r^{1/2}$$

as our initial points. The learning rate $\eta$ was tuned to be $\eta = 10^{-3}$ for the fMRI experiments, and $\eta = 10^{-4}$ for the video experiments. GNMR was used in a similar manner, where we used $\mathbf{Y}$ and $\Omega$ to reconstruct $\mathbf{X}$. We use default values in the code provided by Zilber & Nadler (2022).

Furthermore, we expand on the tangent regression baseline introduced in Section 5.3. To generate this baseline, we perform the following steps:

1. **Reference Estimate:** $\mathbf{U}_0 = $ top-$r$ left singular vectors of $\mathbf{Y}$, i.e. the global ($r$-SVD) subspace.

2. **Local Estimates:** $\widehat{\mathbf{U}}_i = $ top-$r$ left singular vectors of $\mathbf{Y}_i \in \mathbb{R}^{n \times r}$, one per time point.

3. **Lift (log map):** Map each local estimate into the linear tangent space at $\mathbf{U}_0$,

$$\boldsymbol{\xi}_i \;=\; \mathrm{Log}_{\mathbf{U}_0}(\widehat{\mathbf{U}}_i), \qquad \mathbf{U}_0^\top \boldsymbol{\xi}_i = \mathbf{0}.$$

4. **Affine Fit in Tangent Space:** Regress the lifted points on time

$$(\widehat{\mathbf{A}}, \widehat{\mathbf{B}}) = \arg\min_{\mathbf{A}, \mathbf{B}} \sum_{i=1}^{T} \big\| \boldsymbol{\xi}_i - (\mathbf{A} + t_i \mathbf{B}) \big\|_F^2, \qquad \widetilde{\boldsymbol{\xi}}_i = \widehat{\mathbf{A}} + t_i \, \widehat{\mathbf{B}}.$$

5. **Map Back and Project:**

$$\widetilde{\mathbf{U}}_i = \mathrm{Exp}_{\mathbf{U}_0}(\widetilde{\boldsymbol{\xi}}_i), \qquad \widehat{\mathbf{X}}_i^{\mathrm{tan}} = \widetilde{\mathbf{U}}_i \, \widetilde{\mathbf{U}}_i^\top \mathbf{Y}_i.$$

Because $\mathrm{Exp}_{\mathbf{U}_0}(t\widehat{\mathbf{B}})$ traces a geodesic, the linear fit $\widetilde{\boldsymbol{\xi}}_i = \widehat{\mathbf{A}} + t_i\widehat{\mathbf{B}}$ is a first-order approximation of a geodesic through the data: the subspace is now allowed to move, but only along a straight line in the tangent coordinates at $\mathbf{U}_0$. This makes it a natural baseline to compare to our method when $\ell \geq r$, where we still observe that our algorithm yields superior performance.

**Video Experiments.** We provide quantitative results on the mouse dataset with $r = 5$ as well as more frames from both the mouse and plane datasets from Section 5. Echoing from the main paper, Figure 17, shows that the mouse is more clearly present in the frame using our method with higher PSNR. In the bottom row, we can see that the globally low-rank methods show a slight tint of the mouse in a frame where the mouse should be present, whereas it is completely invisible using our algorithm. In Figure 5, we also show another frame of the plane experiments, showing similar observations to the mouse experiments.

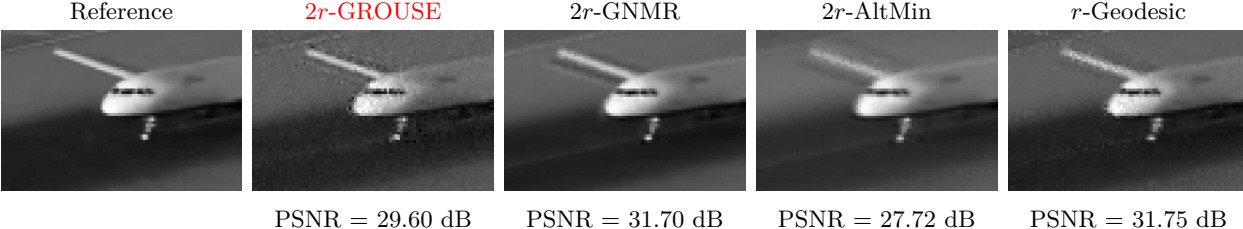

Figure 16: Additional frames of the plane video dataset with $r = 5$. The wing of the plane in the frame using our algorithm is less blurry compared to the globally low-rank methods such as AltMin.

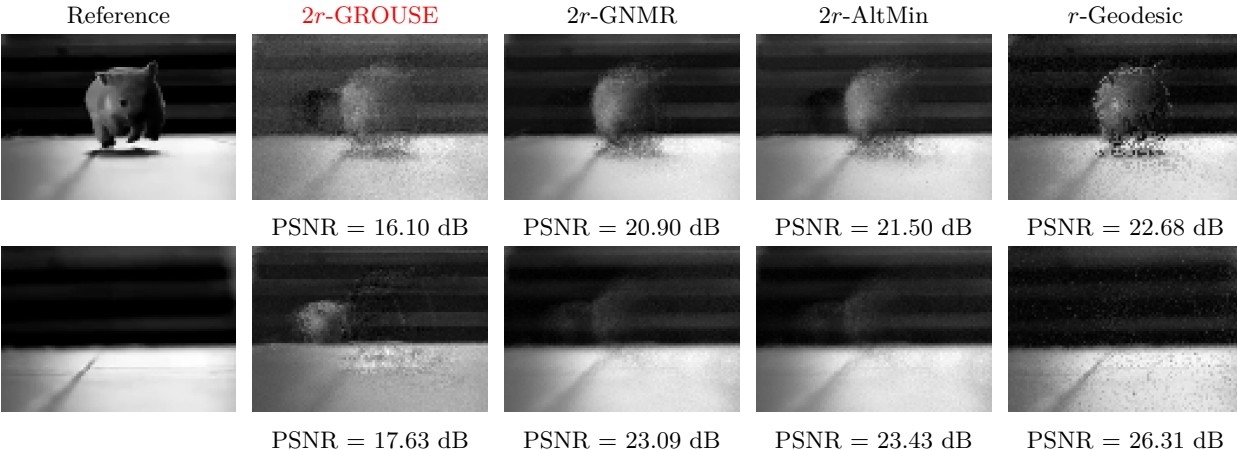

Figure 17: Additional frames from the mouse experiment with $r = 3$ and $\sigma_{\mathbf{y}} = 0.01$. The mouse is clearer using the geodesic method with consistently higher PSNRs.

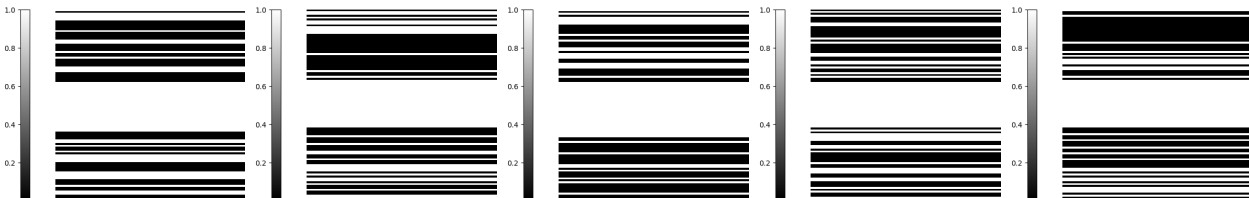

Figure 18: Examples of $k$-space masks, used in both the fMRI data and the simulated MRI data, where approximately 50% of the pixels are zeroed out, i.e., not measured. These specific masks were used to generate the sub-sampled $k$-space data for the OSSI dataset, and the masks for the Shepp Logan Phantoms were generated in a similar manner.

**Dynamic MRI Simulations.** The main text provided results on fMRI reconstruction using the OSSI dataset. In Figure 20, we show the error images, which is the absolute difference between the true and the reconstructed images. Clearly, our method yields a lower error, as there are fewer artifacts across the displayed time points. In Figure 18, we provide examples of what the $k$-space masks look like. While the center of the masks was fully observed, the rows above and below the center were missing, accounting for the percentage of missing pixels. Lastly, Figure 19a displays the singular value plots for the reconstructed data in the undersampled setting (using data sharing) compared to the fully sampled case. We observe that the rank can be effectively approximated from these reconstructions, justifying our choice of rank.

Next, we provide additional results on dynamic 2D Shepp-Logan phantoms (Shepp & Logan, 1974), where the constrasts of the ellipses vary over time. We synthetically generated these phantoms with for a total of

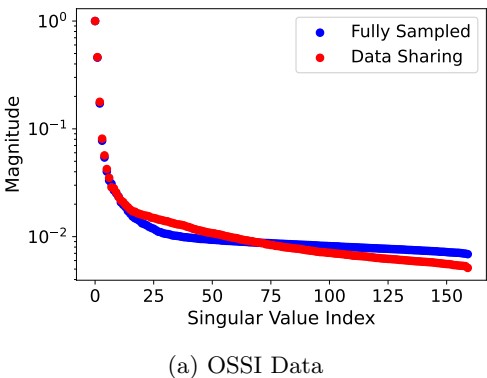

(a) OSSI Data

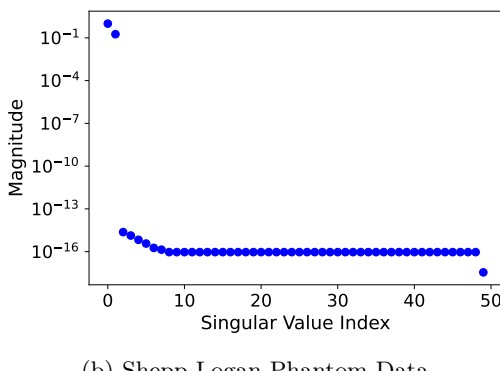

(b) Shepp-Logan Phantom Data

Figure 19: Left: Singular value plot for the OSSI dataset. The rank for the undersampled setting was selected based on data sharing, as it closely approximates the rank of the fully sampled case. Right: Singular value plot for the Shepp-Logan Phantom dataset in the fully sampled case; the data is shown to be exactly $r = 2$.

100 frames, where each frame had dimensions ($100 \times 100$). For the task, we considered the same sub-sampled $k$-space problem as performed in Section 5. We used subspace dimension $r = 2$, where for our algorithm, we chose $\ell = 1$. Since $\ell = 1$, we have a total of $T = 100$ samples. We used evenly spaced time points $t_i \in [0, 1]$ and chose $\lambda = 0.1$. We ran all algorithms until the difference in NRMSE between two successive iterations fell below $10^{-9}$.

Figure 21 shows the results. For both AltMin and our algorithm, we used the nearest-neighbors method as the initial point. As shown in Figure 19b, the Shepp-Logan phantom images form an exact low-rank matrix with $r = 2$. Therefore, we do not expect our method to outperform AltMin; instead, our goal was to asses whether the geodesic structure can still provide accurate recovery for dynamic MRI tasks. Figure 21 confirms our hypothesis, as we are able to reconstruct the MRI images closely to the AltMin baseline.

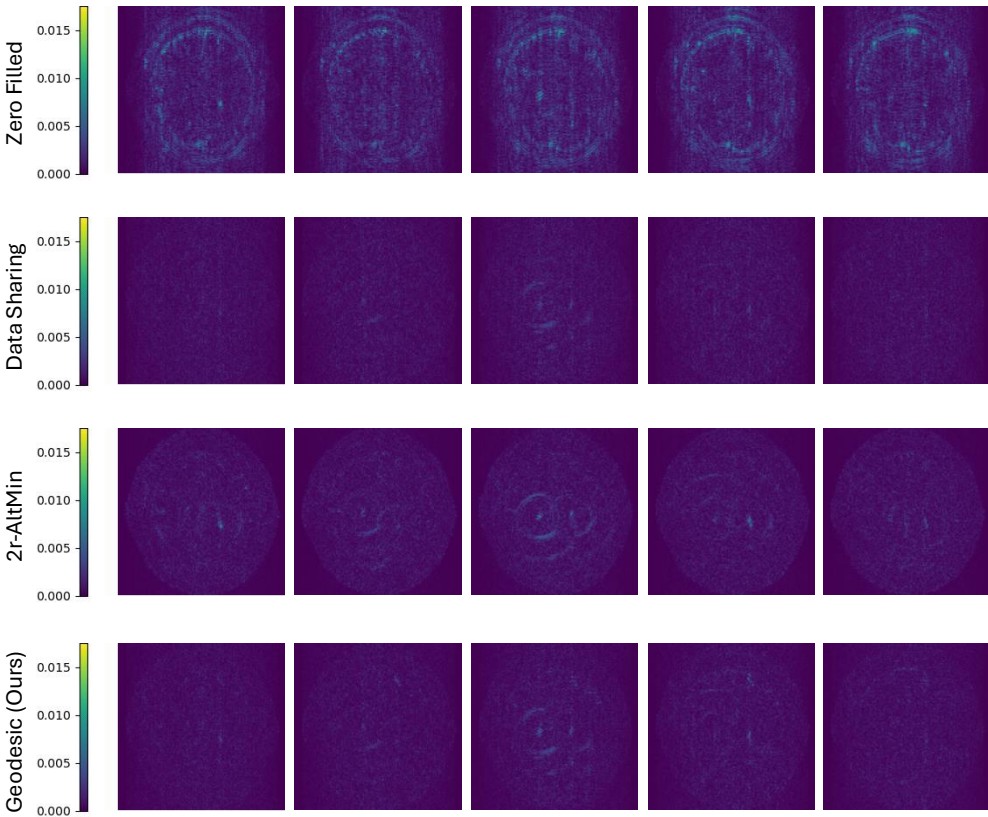

Figure 20: Depiction of the absolute difference between the reconstructed and true images in Figure 8. Since our method achieves a lower NRMSE across all displayed time points, it also exhibits fewer artifacts in the error images.

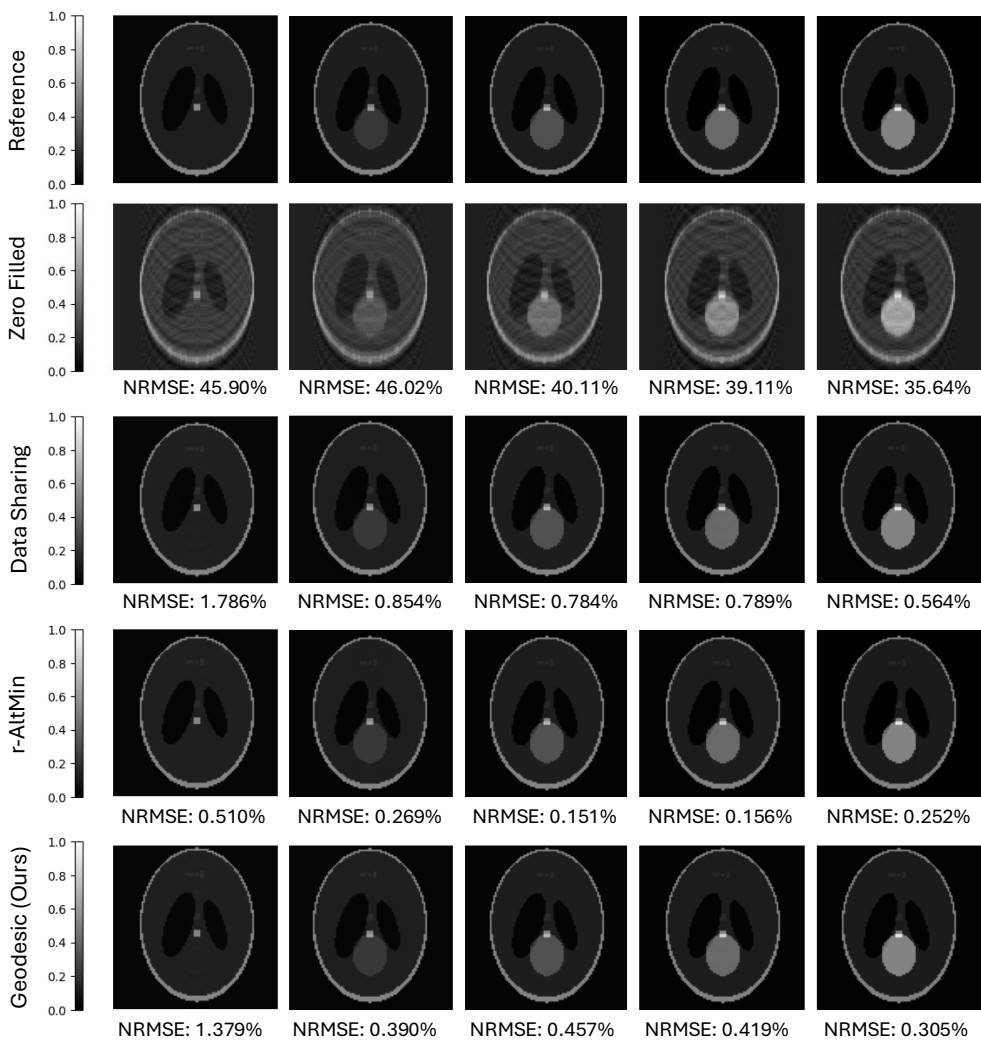

Figure 21: Results on MRI reconstruction of dynamic 2D Shepp-Logan phantoms (Shepp & Logan, 1974), where we observe approximately 50% of the *k*-space of each frame. Compared to zero-filled reconstruction, the proposed Geodesic approach obtained lower d values.

## C  Deferred Proofs

To keep this section self-contained, we first restate all of our results.

**Theorem 1** (Convergence to a Stationary Point). *Let $\mathbf{\Psi}^k = \{\mathbf{X}_i^k\}_{i=1}^T \cup \{\mathbf{Q}^k, \mathbf{\Theta}^k\}$ denote the iterates generated by Algorithm 1 at iteration $k$ starting from any initialization with proximal parameters $\lambda_{\mathbf{x}}, \lambda_{\mathbf{Q}}, \lambda_{\mathbf{\Theta}} > 0$. For any $\epsilon > 0$, if $k = \widetilde{\mathcal{O}}(\epsilon^{-2})$, then we have*

$$\sum_{p=1}^{T+2} \|\nabla f_{(p)}(\mathbf{\Psi}_p^k)\| \leq \epsilon, \tag{34}$$

*where $p$ indexes the $T + 2$ sequentially updated blocks of variables in $\mathbf{\Psi}^k$.*

*Proof.* Let $f$ denote the objective function in Equation (4), $f_{(p)}^k$ denote the marginal loss with respect to the $p$-th block at iteration $k$, and $g_{(p)}^k$ be a majorizer for $f_{(p)}^k$. Following the notation in the main text, we often equivalently write the marginal as its argument when it is clear from the context (e.g., $f^k(\mathbf{x}_{i,j}) = f_{(\mathbf{x}_{i,j})}^k$). We can invoke (Li et al., 2023, Corollary 3.7) (restated below as Theorem 3) by showing the following:

- (Quadratic Majorization Gap). $g_{(p)}^k(\omega) - f_{(p)}^k(\omega) \geq c\|\omega - \omega_{(p)}^{k-1}\|^2$, $\forall p$, for some constant $c > 0$

- (Smoothness of Marginal Loss). $\|\nabla_{\omega_1} f_{(p)}^k(\omega_1) - \nabla_{\omega_2} f_{(p)}^k(\omega_2)\| \leq L\|\omega_1 - \omega_2\|$, $\forall p$

- (Smoothness of Surrogate Loss). $\|\nabla_{\omega_1} g_{(p)}^k(\omega_1) - \nabla_{\omega_2} g_{(p)}^k(\omega_2)\| \leq L\|\omega_1 - \omega_2\|$, $\forall p$

**Quadratic Majorization Gap.** By construction, Algorithm 1 introduces explicit proximal parameters $\lambda_x, \lambda_Q, \lambda_\Theta > 0$ when formulating the majorizer for each respective block. These proximal terms inherently enforce a strict quadratic gap between the surrogate and the marginal loss, which satisfy this condition.

**Smoothness of Marginal Loss.** Firstly, we show that the marginal loss with respect to each data matrix $\mathbf{X}_i$ is $L_{\mathbf{x}}$-smooth by bounding the spectral norm of its Hessian. For each data point $\mathbf{x}_{i,j}$, we have the marginal loss

$$f(\mathbf{x}_{i,j}) = \frac{1}{2} \left\| \widetilde{\mathbf{y}}_{i,j} - \widetilde{\mathbf{A}}_{i,j} \mathbf{x}_{i,j} \right\|_2^2.$$

The second directional derivative for any arbitrary vector $\mathbf{\Delta} \in \mathbb{R}^n$ is:

$$\nabla^2 f(\mathbf{x}_{i,j})[\mathbf{\Delta}, \mathbf{\Delta}] = \left\| \widetilde{\mathbf{A}}_{i,j} \mathbf{\Delta} \right\|_2^2.$$

Applying the definition of the induced $L_2$ operator norm, we can strictly bound this curvature:

$$\left\| \widetilde{\mathbf{A}}_{i,j} \mathbf{\Delta} \right\|_2^2 \leq \left\| \widetilde{\mathbf{A}}_{i,j} \right\|_2^2 \|\mathbf{\Delta}\|_2^2.$$

Thus, the marginal loss for each $\mathbf{x}_{i,j}$ is globally smooth with Lipschitz constant $L_{\mathbf{x}} = \max_{i,j} \left\| \widetilde{\mathbf{A}}_{i,j} \right\|_2^2$.

Next, we consider Euclidean smoothness of the block $\mathbf{Q}$. The marginal loss with respect to $\mathbf{Q}$ is given by $f(\mathbf{Q}) = -\frac{\lambda}{2} \sum_{i=1}^T \text{tr}\left(\mathbf{Q}^\top \mathbf{X}_i \mathbf{X}_i^\top \mathbf{Q} \mathbf{R}_i \mathbf{R}_i^\top\right)$. With a slight abuse in notation, for an arbitrary matrix direction $\mathbf{\Delta} \in \mathbb{R}^{n \times 2r}$, the second directional derivative is

$$\nabla^2 f(\mathbf{Q})[\mathbf{\Delta}, \mathbf{\Delta}] = -\lambda \sum_{i=1}^T \text{tr}\left(\mathbf{\Delta}^\top \mathbf{X}_i \mathbf{X}_i^\top \mathbf{\Delta} \mathbf{R}_i \mathbf{R}_i^\top\right).$$

Because both $\mathbf{\Delta}^\top \mathbf{X}_i \mathbf{X}_i^\top \mathbf{\Delta}$ and $\mathbf{R}_i \mathbf{R}_i^\top$ are positive semi-definite matrices, we apply the trace inequality $\text{tr}(\mathbf{AB}) \leq \|\mathbf{B}\|_2 \cdot \text{tr}(\mathbf{A})$:

$$\left|\text{tr}\left(\mathbf{\Delta}^\top \mathbf{X}_i \mathbf{X}_i^\top \mathbf{\Delta} \mathbf{R}_i \mathbf{R}_i^\top\right)\right| \leq \left\|\mathbf{R}_i \mathbf{R}_i^\top\right\|_2 \cdot \text{tr}\left(\mathbf{\Delta}^\top \mathbf{X}_i \mathbf{X}_i^\top \mathbf{\Delta}\right).$$

Since $\mathbf{R}_i^\top \mathbf{R}_i = \mathbf{I}_r$, the non-zero eigenvalues of $\mathbf{R}_i \mathbf{R}_i^\top$ are strictly 1, meaning $\left\| \mathbf{R}_i \mathbf{R}_i^\top \right\|_2 = 1$. Then, we bound the remaining trace using the Frobenius norm and the spectral norm of $\mathbf{X}_i$:

$$\mathrm{tr}\left( \boldsymbol{\Delta}^\top \mathbf{X}_i \mathbf{X}_i^\top \boldsymbol{\Delta} \right) = \left\| \mathbf{X}_i^\top \boldsymbol{\Delta} \right\|_\mathsf{F}^2 \leq \left\| \mathbf{X}_i \right\|_2^2 \cdot \left\| \boldsymbol{\Delta} \right\|_\mathsf{F}^2.$$

Combining these inequalities yields:

$$\left| \nabla^2 f_\mathbf{Q}(\mathbf{Q})[\boldsymbol{\Delta}, \boldsymbol{\Delta}] \right| \leq \left( \lambda \sum_{i=1}^T \left\| \mathbf{X}_i \right\|_2^2 \right) \left\| \boldsymbol{\Delta} \right\|_\mathsf{F}^2.$$

This establishes global $L_\mathbf{Q}$-smoothness for $\mathbf{Q}$ with the precise Lipschitz parameter $L_\mathbf{Q} = \lambda \sum_{i=1}^T \left\| \mathbf{X}_i \right\|_2^2$.

Lastly, we need to verify the smoothness of the marginal loss with respect to each coordinate $\theta_s$. Recall that the derivative is given by $\dot{f}_{i,s}(\theta_s) = \lambda r_{i,s} t_i \sin(2\theta_s t_i - \phi_{i,s})$. Because this is a 1D function, we establish rigorous smoothness by directly computing the supremum of the absolute second derivative:

$$\begin{aligned}|\ddot{f}_{i,s}(\theta_s)| &= \left| 2\lambda r_{i,s} t_i^2 \cos(2\theta_s t_i - \phi_{i,s}) \right| \\ &\leq 2\lambda r_{i,s} t_i^2.\end{aligned}$$

Thus, the marginal loss is smooth with parameter $L_\Theta = 2\lambda \sum_{i=1}^T r_{i,s} t_i^2$.

**Smoothness of Surrogate Loss.** The constructed surrogate functions $g_{(p)}^k$ for all variable blocks are exactly quadratic with respect to their active variables. Because quadratic functions inherently possess constant Hessian matrices, their gradients are uniformly Lipschitz continuous, thereby satisfying Euclidean $L'$-smoothness globally.

This completes the proof.

$\square$

**Theorem 2** (Monotonic Objective Function). *Let $\boldsymbol{\Psi}^k = \{\mathbf{X}_i^k\}_{i=1}^T \cup \{\mathbf{Q}^k, \boldsymbol{\Theta}^k\}$ denote the iterates generated by Algorithm 1 at iteration $k$. Algorithm 1 produces blocks that are monotonically non-increasing in the loss: $f(\boldsymbol{\Psi}^k) \geq f(\boldsymbol{\Psi}^{k+1})$.*

*Proof.* We establish monotonicity by validating the MM descent principle for each iteratively updated block (Sun et al., 2017). That is, for a generic block $\omega$ with current iterate $\omega^k$ and marginal objective $\varphi(\omega)$, a valid surrogate $g^k(\omega; \omega^k)$ must satisfy the following:

$$g^k(\omega^k; \omega^k) = \varphi(\omega^k) \tag{Tangency}$$

$$g^k(\omega; \omega^k) \geq \varphi(\omega). \tag{Domination}$$

With these two properties, if $\omega^{k+1} = \arg\min_\omega g^k(\omega; \omega^k)$, strict non-increase is guaranteed:

$$\varphi(\omega^{k+1}) \leq g^k(\omega^{k+1}; \omega^k) \leq g^k(\omega^k; \omega^k) = \varphi(\omega^k).$$

We validate these two properties for each block.

**X Verification.** From the second-order expansion with the proximal term (Appendix A), the marginal admits the quadratic upper bound

$$f(\mathbf{x}_{i,j}) \leq \underbrace{\frac{L_\mathbf{x} + \lambda_\mathbf{x}}{2} \left\| \mathbf{x}_{i,j} - \left( \mathbf{x}_{i,j}^k - \frac{1}{L_\mathbf{x} + \lambda_\mathbf{x}} \nabla f(\mathbf{x}_{i,j}^k) \right) \right\|_2^2}_{=:g^k(\mathbf{x}_{i,j}; \mathbf{x}_{i,j}^k)} + \underbrace{\left( f(\mathbf{x}_{i,j}^k) - \frac{1}{2(L_\mathbf{x} + \lambda_\mathbf{x})} \| \nabla f(\mathbf{x}_{i,j}^k) \|_2^2 \right)}_{=:c},$$

where $c$ is a constant independent of $\mathbf{x}_{i,j}$. This satisfies both MM conditions: (i) domination, since the inequality holds for all $\mathbf{x}_{i,j}$ ($g^k(\mathbf{x}_{i,j}; \mathbf{x}_{i,j}^k) + c \geq f(\mathbf{x}_{i,j})$); and (ii) tangency, since evaluating the surrogate at $\mathbf{x}_{i,j} = \mathbf{x}_{i,j}^k$ recovers the marginal, $g^k(\mathbf{x}_{i,j}^k; \mathbf{x}_{i,j}^k) + c = f(\mathbf{x}_{i,j}^k)$.

**Q Verification.** Recall that the marginal is $f(\mathbf{Q}) = \frac{\lambda}{2} \sum_{i=1}^{T} \|(\mathbf{QR}_i\mathbf{R}_i^\top\mathbf{Q}^\top - \mathbf{I}_n)\mathbf{X}_i\|_\mathsf{F}^2$. On the Stiefel manifold, $\mathbf{Q}^\top\mathbf{Q} = \mathbf{I}_{2r}$ and $\mathbf{R}_i^\top\mathbf{R}_i = \mathbf{I}_r$ make $\mathbf{QR}_i\mathbf{R}_i^\top\mathbf{Q}^\top$ idempotent, and so the marginal reduces to

$$f(\mathbf{Q}) = \frac{\lambda}{2}\sum_{i=1}^{T}\|\mathbf{X}_i\|_\mathsf{F}^2 - \frac{\lambda}{2}\sum_{i=1}^{T}\|\mathbf{X}_i^\top\mathbf{QR}_i\|_\mathsf{F}^2, \qquad \forall\,\mathbf{Q}\in\mathcal{V}^{n\times 2r},$$

which is concave in $\mathbf{Q}$. The surrogate (Appendix A) linearizes this reduction at $\mathbf{Q}^k$ and adds a proximal term,

$$g^k(\mathbf{Q}) := f(\mathbf{Q}^k) + \langle\nabla f_\mathbf{Q}(\mathbf{Q}^k),\, \mathbf{Q}-\mathbf{Q}^k\rangle \underbrace{+\frac{\lambda_\mathbf{Q}}{4}\|\mathbf{Q}-\mathbf{Q}^k\|_2^2}_{\text{to ensure quadratic gap}}. \tag{35}$$

By concavity, the linearization lies above $f$ on the manifold, and the proximal term is nonnegative; hence

$$f(\mathbf{Q}^k) = g^k(\mathbf{Q}^k) \quad\text{and}\quad f(\mathbf{Q}) \le g^k(\mathbf{Q}) \quad \forall\mathbf{Q}\in\mathcal{V}^{n\times 2r}.$$

**$\Theta$ Verification.** For a coordinate $s$, the marginal $f_{(\theta_s)}(\theta_s) = \sum_{i=1}^{T} f_{i,s}(\theta_s)$ has bounded curvature $|\ddot{f}_{i,s}(\theta_s)| \le 2\lambda r_{i,s}t_i^2$. The algorithm constructs the quadratic surrogate

$$g_{i,s}(\theta_s;\theta_s^k) = f_{i,s}(\theta_s^k) + \dot{f}_{i,s}(\theta_s^k)(\theta_s-\theta_s^k) + \frac{1}{2}w_{f_{i,s}}(\theta_s^k)(\theta_s-\theta_s^k)^2,$$

whose curvature satisfies $w_{f_{i,s}}(\theta_s^k) \ge 2\lambda r_{i,s}t_i^2$. Tangency holds by construction, and since the curvature dominates $|\ddot{f}_{i,s}|$, the descent lemma yields domination: $f_{(\theta_s)}(\theta_s) \le \sum_{i=1}^{T} g_{i,s}(\theta_s;\theta_s^k)$.

**Chaining the Updates.** Because each block update minimizes a dominating surrogate while holding all other elements strictly fixed, sequential applications yield the following:

$$\begin{aligned}
f(\mathbf{\Psi}^k) &= f(\mathbf{X}^k, \mathbf{Q}^k, \mathbf{\Theta}^k)\\
&\ge f(\mathbf{X}^{k+1}, \mathbf{Q}^k, \mathbf{\Theta}^k)\\
&\ge f(\mathbf{X}^{k+1}, \mathbf{Q}^{k+1}, \mathbf{\Theta}^k)\\
&\ge f(\mathbf{X}^{k+1}, \mathbf{Q}^{k+1}, \mathbf{\Theta}^{k+1}) = f(\mathbf{\Psi}^{k+1}).
\end{aligned}$$

This establishes monotonic non-increase of the full objective function over each iteration $k$. This completes the proof.

$\square$

## D  Auxiliary Results

**Definition 1** (Optimality Gap). *Let $f$ be a smooth $p$-block function $f : \Omega_{(1)}\times\ldots\times\Omega_{(p)} \to \mathbb{R}$ with majorizing surrogate $g_{(i)}^k(\omega)$ for the $k$-th iteration and with respect to the $i$-th block. The optimality gap is defined as*

$$\Delta_k := \max_{1\le i\le p}\left(g_{(i)}^k(\omega^k) - \inf_\omega g_{(i)}^k(\omega)\right).$$

**Definition 2** (Inexact Computation). *Let $g^k(\omega;\omega^k)$ denote the majorizing surrogate at iteration $k$ to the function $f(\omega)$. We say that a majorizer $g^k$ is inexactly computed if*

$$\|\omega^{k+1} - \omega^{k\star}\|_2 = \mathcal{O}(1),$$

*where $\omega^{k+1}$ is an output of the majorizer $g^k(\omega;\omega^k)$ and $\omega^{k\star}$ is an exact solution to $g^k(\omega;\omega^k)$.*

Notice that the update steps in Algorithm 1 require finding a minimizer to a surrogate function. We will use Definition 2 to allow inexact computation in our proof for convergence to a stationary point.

**Theorem 3** (Complexity of RBMM on Stiefel Manifolds; Corollary 3.7 (Li et al., 2023)). *Consider the minimization of a smooth p-block function $f : \Omega_{(1)} \times \ldots \times \Omega_{(p)} \to \mathbb{R}$ via RBMM, where each underlying manifold $\Omega_{(i)}$ is either a Stiefel manifold or a Euclidean space. Suppose that $f$ is Euclidean $L$-smooth for some $L > 0$. Suppose that the surrogates $g_{(i)}^k$ at iteration $k$ are Euclidean $L'$-smooth for some constants $L' > 0$ and for some constant $c > 0$,*

$$g_{(i)}^k(\omega) - f_{(i)}^k(\omega) \geq c\|\omega - \omega_{(i)}^{k-1}\|_2^2,$$

*for all $k \geq 1$ and $i \in [p]$, where $f_{(i)}^k$ is the marginal objective function of $f$ at iteration $k$. Allow inexact computation of the surrogate functions at each iteration in the sense of Definition 2 and assume that the optimality gaps are summable (i.e. $\sum_{k=1}^{\infty} \Delta_k < \infty$). Then, the iterations produced by any RBMM algorithm starting from random initialization asymptotically converge to the set of stationary points with an iteration complexity of $\widetilde{\mathcal{O}}(\epsilon^{-2})$.*

