# OpenReview forum: "Dynamic Subspace Estimation from Undersampled Data using Grassmannian Geodesics"
_TMLR — Decision pending for TMLR_

### Review · Reviewer_Bbv6 · 2026-05-28

**Summary Of Contributions:**

The manuscript proposes a Riemannian block majorize-minimization method to estimate a sequence of low-dimensional subspaces from possibly undersampled data. The sequence of subspaces is lying on a geodesic on the Grassmann manifold, and the proposed method estimates this geodesic together with a data matrix coming from each subspace. The authors prove that the proposed algorithm convergences to a stationary point. A number of experiments shows the efficiency of the algorithm. The synthetic experiments demonstrate the advantages of the proposed method when the data is generated from subspaces evolving along a geodesic. This is not particularly surprising, since the method is evaluated under the same assumption used to formulate the problem. A more important question is whether the geodesic assumption is realistic or useful in practical applications. The additional experiments help address this point.

**Audience:**

Yes

**Audience Explanation:**

I believe these findings will be particularly interesting to researchers working in video processing.

**Broader Impact Concerns:**

I do not identify significant broader impact concerns beyond those generally associated with machine learning methods.

**Claims And Evidence:**

Yes

**Claims Explanation:**

The authors provide detailed proofs of the update steps and the convergence results in the supplementary material.

**Requested Changes:**

The manuscript is clearly written and well structured. I only have a few minor comments below.

1) The data model subsection in the problem formulation is a bit unclear. The problem statement and the motivation is mixed. I suggest posing the problem statement clearly first (the first two equations) and then motivating the problem setup.

2) Please discuss the relation between your method and GROUSE. That algorithm also follows a geodesic on the Grassmannian. I suppose the main difference is that they follow a different geodesic at each update step towards the newly observed data, and the proposed framework fits a single geodesic to the whole dataset.

3) The function $f$ is multiply defined. As a consequence, the main statement in Theorem 1 is not clear without looking at the appendix. Please state explicitly what f is in Theorem 1.

4) For the initialization of $H$, $Z$, and $\Theta$: why do you compute $H_1$ from $X_1$ and $x_{2,1}$? This choice seems as hoc, and also seem to require that $r = \ell + 1$.

---

> ### Author Response · Authors · 2026-06-12
> **Author Response**
>
> Thank you for taking the time to review our paper. Below, we address the concerns in detail:
>
> > **"The data model subsection in the problem formulation is a bit unclear. The problem statement and the motivation is mixed."**
>
> **Response:** We have revised the manuscript so that the data model section focuses solely on the generative model formulation and the partially observed measurements. If you believe this section requires further clarification, we would be happy to further refine it.
>
>
> > **"Please discuss the relation between your method and GROUSE."**
>
> **Response:** GROUSE indeed follows a geodesic for its stochastic gradient update, as would any manifold gradient descent algorithm, and that geodesic changes for each gradient descent update. One motivating reason for studying the batch geodesic learning problem was precisely that methods like GROUSE are actually optimizing a fixed objective function in a streaming/online way, and the streaming aspect of the algorithm is often co-opted to handle (slowly) time-varying data. We took a step back and asked what a natural time-varying model was, and considered the time-varying subspace as a curve on the Grassmannian. Of course (as has been pointed out elsewhere in the review), fitting an arbitrary curve would be challenging; a piecewise geodesic approximation is a natural approach, and the first step to that is understanding how to optimize data-fit to a single geodesic. Part of future work is to study a streaming/online algorithm that would solve our optimization formulation, including adding piecewise geodesics.
>
> > **"The function f is multiply defined."**
>
> **Response:** Thank you for raising this point. We have revised the manuscript such that the function $f$ refers to the overall objective, and each marginal is denoted as $f_{(p)}$ for the $p$-th block. However, when the active block is clear from the context, we equivalently write the marginal as its argument (e.g., $f(\mathbf{x}_{i, j})$). This is specified in the beginning of Section 3.1.
>
> >  **"why do you compute $\mathbf{H}_1$... this choice seems as hoc, and also seem to require that $r=\ell + 1$"**
>
> **Response:**
> - In Equation (17), we used these for initialization to address scenarios where the number of available data points at a given time step (e.g., in the data matrix $\mathbf{X}_1$) is strictly less than the chosen rank. If the rank is smaller than $\ell$, we can simply use $\mathbf{X}_1$. We emphasize that our algorithm does not require $r = \ell + 1$; rather, we chose $r = \ell + 1$ in Section 5.1 specifically to demonstrate that our method can robustly handle cases where $\ell < r$.
>
> - However, this setting naturally raises a related question: how does our algorithm perform compared to alternative geodesic baseline algorithms when $\ell > r$? To address this (a point also raised by another reviewer), we have added an ablation study in Section 5.3. For this regime, one can estimate a subspace independently at each time point by computing the top-$r$ left singular vectors of the noisy observation matrix. This yields a natural baseline: (i) estimate the local subspaces, (ii) project them to the tangent space at a reference subspace $\mathbf{U}_{\text{ref}}$ via the logarithmic map, (iii) fit an affine trajectory $\hat{\mathbf{A}} + t_i \hat{\mathbf{B}}$ in this tangent space, and (iv) map the fitted tangent vectors back to the Grassmannian using the exponential map. In Section 5.3, we show that our algorithm consistently outperforms this baseline, further validating our approach.

---

### Review · Reviewer_ayvE · 2026-05-30

**Summary Of Contributions:**

The authors propose a method to identify dynamic subspaces from undersampled measurements. The main idea is to model the time-varying subspaces as points on a Grassmannian geodesic (as a form of regularization). In doing so, the method opts for an alternative optimization approach and also formulates steps of optimization as majorization-minimization.

I believe the main strength of the paper is its geometric approach. The main weaknesses are discussed in the sections below.

**Additional Comments:**

## Minor Comments

- The caption of Figure 1 is unclear. The phrase “$H_1,\dots,H_4$ are orthonormal bases for a point on the Grassmannian” should be revised. Each orthonormal basis represents a subspace, i.e. a point on the Grassmannian.

- In the notation section, please explicitly mention $r \leq n$.

- In Section 2.1, please clarify what is meant by a “non-trivial null space.”

- The statement “the latent signals are smoothly related over time in such a way that their low-rank approximations are also smoothly related” needs references.

- Please clarify the role of $\ell$, the number of samples per time point. In some experiments it seems tied to $r$, e.g. $\ell=r-1$. Is this only an experimental choice, or is there a modelling reason?

- Please define $U$ around Eq. (2).

- Regarding the remark underneath eq.5, can we just simply assume the regularizer deems $x_{i,j}$ not to have any value in the orthogonal complement of the $U_i$, hence it is a good regularizer (of course I would write it as $I _ U_iU_i^T$)?

- In Algorithm 1, please explicitly state how $B_{i,j}$ is constructed.

- In Eq. (5), $Y$ is the observed measurement and is given. Presumably the authors meant to optimise over $Z$, not $Y$.

- In Section 2.1, the phrase “if $A_{i,j}$ are subsampled diagonal matrices” is unclear. The dimensionality of $A_{i,j}$ is ${m\times n}$. Please clarify what the subsampled diagonal means here.

- The identity-map discussion in Section 2.2 is dimensionally ambiguous. In Eq. (2), $Y\in\mathbb{R}^{m\times \ell T}$, while $UG\in\mathbb{R}^{n\times \ell T}$.

- As mentioned before, I found the treatment of the time parameters $t_i$ somewhat unclear. In the problem formulation, the time points appear to be treated as given inputs to the model, with the subspaces parameterised in the constraint of eq.5. Similarly, in the pseudo-code of the algorithm, $t_i$ is given. However, in the experiments, the choice of the interval appears to be tuned in a dataset-dependent way. Is it correct to assume that $t_i$ is an additional set of hyperparameters of the proposed model?
- related to the above, in a practical setting, how $t_i$ should be chosen?
- Again related to $t_i$, I think an ablation on the effect of this hyperparameter is required.
- I found the rank comparison somewhat incomplete. In the proposed method the subspaces are parameterised through H and Z, so the model effectively uses a shared ambient subspace of dimension up to 2r. The paper therefore compares against both rank-r and rank-2r static baselines, which is reasonable as a first comparison. However, the experiments do not report what happens when the static baselines are allowed to use ranks larger than 2r, or when their rank is selected by a validation protocol. To me this is needed as the gap between the proposed method and a static method with rank 2r in some cases is not tangible.

**Audience:**

Yes

**Audience Explanation:**

The problem considered in the paper has both theoretical and practical relevance. Subspace estimation and inverse problems are significant across many topics.

**Broader Impact Concerns:**

I do not see any major broader-impact concerns here

**Claims And Evidence:**

No

**Claims Explanation:**

Please see my comments below.

**Requested Changes:**

I will add my major concerns here and relegate the minor ones to the comment section below.
## Major Comments

1. The proposed method constrains the time-varying subspaces to lie on a single Grassmannian geodesic. This is interesting, but seems restrictive, since a smooth trajectory of subspaces need not be geodesic. There are also some concerns related to proposed formulation here. In Algorithm 1, the time parameters $t_i$ are considered as inputs. However, in the experiments, $t_i$ is chosen from an interval. This mismatch could significantly change the performance of the model.

I also think a natural baseline is missing. One could first estimate local subspaces $\widehat U_i \in \mathrm{Gr}(r,n)$, map them to the tangent space of a reference subspace $U_0$ using the Grassmann logarithm map,
\[
\xi_i = \operatorname{Log}_{U_0}(\widehat U_i),
\]
fit a spline or other smooth curve in the tangent space, and then map back using the exponential map.


2. I found the empirical evaluation somewhat limited. A substantial part of the synthetic evaluation uses data generated from the same geodesic assumption used by the model. This is useful as a sanity check, but it does not test robustness to model misspecification or to smooth-but-non-geodesic dynamics. The real-data experiments on video and fMRI are also fairly narrow. The videos are low-resolution, and the comparisons are mostly against static low-rank baselines. The fMRI experiment also reads more like a proof of concept than a competitive modern reconstruction benchmark.

3. It is also not clearly justified why majorize-minimize is needed for some of the steps of the algorithm. In particular, why the $X$-update needs it as the $x_{i,j}$ subproblem is simply a regularised least-squares problem with closed-form solution. Furthermore, to me to perform the majorization, the Lipschitz constant $L$ constant should be greater than the spectral norm of $\widetilde A_{i,j}$, shouldn’t it? My reading from the text is that this is fixed across all $\widetilde A_{i,j}$ which I cannot fully grasp.

---

> ### Author Response · Authors · 2026-06-12
> **Author Response (Part 1)**
>
> Thank you for taking the time to review our paper. Below, we address the concerns in detail:
>
> > **“I also think a natural baseline is missing. One could first estimate local subspaces…”**
>
> **Response:** Thank you for raising this point. As noted in our global response, we have included this as a 'tangent regression' case study baseline in Section 5.3 for the setting where $\ell \geq r$ and local subspaces can be estimated. The specific steps for this baseline are detailed in Appendix B.2 (and provided in the attached [screenshot](https://imgur.com/a/mgeeQcc)). Even in this regime, our experiments demonstrate that our modeling approach achieves superior performance across all frames in terms of PSNR on a video dataset. The only necessary design choice for this baseline is the estimation of the global (or reference) subspace; we approximated this using the top left singular vectors of the concatenated data, which we believe is a natural approach. Finally, we emphasize that tangent regression is strictly limited to cases where $\ell \geq r$, whereas our proposed method requires no such restriction.
>
> > **“This is interesting, but seems restrictive, since a smooth trajectory of subspaces need not be geodesic … In Algorithm 1, the time parameters $t_i$ are considered as inputs. However, in the experiments, $t_i$ is chosen from an interval.”**
>
> **Response:**
> - We agree that a smooth trajectory of subspaces need not be geodesic. However, we note that existing methods such as GROUSE also use geodesics in their stochastic gradient updates, as does any manifold gradient descent algorithm. In these methods, the geodesic is primarily used as an optimization tool: each update follows a different geodesic determined by the current gradient direction.
> 0 One motivating reason for our modeling assumption is that methods such as GROUSE optimize a fixed objective function in a streaming or online manner, and the streaming nature of the algorithm is often used to handle slowly time-varying data. We instead take a step back and ask what a natural model for time-varying subspaces should be. This leads us to model the time-varying subspace as a curve on the Grassmannian. Of course, fitting an arbitrary curve would be challenging; a piecewise-geodesic approximation is a natural approach, and the first step toward this goal is to understand how to optimize a data-fitting objective over a single geodesic, which is the main focus and contribution of this work. Our experiments, including the baseline suggested by the reviewer, support this modeling assumption.
> - The time parameters are treated as hyperparameters and are therefore inputs to the algorithm; they should be chosen from a prescribed interval. A natural choice is to sample them evenly from $t_i \in [0,1]$, so that $\mathbf{U}_0$ corresponds to the initial subspace generated from $\mathbf{X}_0$ and $\mathbf{U}_T$ corresponds to the subspace generated from $\mathbf{X}_T$. However, as we show in the newly added ablation study in Section 5.3, choosing interior points (e.g., sampled time steps $t_i \in [0.3,0.7]$) often yields better results, which motivates our specific choice of timesteps.
>
> > **“I found the empirical evaluation somewhat limited… The fMRI experiment also reads more like a proof of concept than a competitive modern reconstruction benchmark.”**
>
> **Response:**
> - We agree that there may be applications outside of video reconstruction that better reflect the geodesic modeling assumption. However, our experimental results demonstrate the promise of this assumption even on video datasets, where we do not expect the underlying subspaces to lie exactly on geodesics. The fact that our method outperforms the baselines considered in the manuscript suggests that geodesic modeling is a promising avenue for future research.
> - For the fMRI data, most dynamic MRI settings that employ low-rank models assume a single subspace, either for the entire time series or for the time series of each patch. We focus on the specific and specialized OSSI application because it has two time indices, slow time and fast time, and the MRI physics suggests that the signal manifold, which we approximate here by a subspace, should evolve slowly over time. This makes OSSI a natural application for the proposed geodesic method. To our knowledge, there are no established benchmarks for the OSSI application. We note that the “data sharing” method we compare against is a *dynamic* reconstruction method and is therefore a natural baseline for this application. The fact that our algorithm outperforms this baseline further supports the use of our method.

---

> ### Author Response · Authors · 2026-06-12
> **Author Response (Part 2)**
>
> > **“It is also not clearly justified why majorize-minimize is needed for some of the steps of the algorithm.”**
>
> **Response:**
> - For the algorithm to be a proper RBMM algorithm satisfying Theorem 2, each block update must minimize a corresponding majorizer. This is why we also construct a majorizer for the data matrix update $\mathbf{X}$, where we include a quadratic gap to ensure that the conditions of Theorem 1 hold. For the Lipschitz constant, we can take the spectral norm of $\widetilde{\mathbf{A}}_{i,j}$ in the update step.
> - Following the suggestion of Reviewer 7QYn, we also clarified in the proof of Theorem 2 that each surrogate satisfies the required majorization conditions.
>
> > **“Regarding the remark underneath eq.5, can we just simply assume the regularizer deems $x_{i, j}$ not to have any value in the orthogonal complement”**
>
> **Response:** The point of our remark here is to give a slightly more precise characterization, arguing why the $\widetilde{\mathbf{A}}$ matrix would be full rank with known subspaces $\mathbf{U}$. To clarify, we changed the wording “Here, we provide some intuition as to why the subspace regularizer may be useful” to “Here, we provide an argument for why this regularization will provide a well-posed optimization problem, despite having very few linear measurements of each vector.”
>
> > **“Please clarify what the subsampled diagonal means here.”**
>
> **Response:** Note that in the missing data case, the dimensions of the observations are the same as the underlying data: $\mathbf{y} = \mathbf{Ax} \in \mathbb{R}^n$ where $\mathbf{x} \in \mathbb{R}^n$. When A is the identity matrix, then $\mathbf{y}=\mathbf{x}$, but if $\mathbf{A}$ is a sub-sampled diagonal matrix (i.e., some diagonal entries of the identity matrix are zero), then we have missing pixels, which is exactly the matrix completion setting.
>
> > **“As mentioned before, I found the treatment of the time parameters t_i somewhat unclear”**
>
> **Response:** We included an ablation study on the time parameters in Section 5.3 that we hope will help clarify.
>
> > **“Please clarify the role of $\ell$, the number of samples per time point. Is this only an experimental choice, or is there a modelling reason?”**
>
> **Response:** In real applications, $\ell$ may be given by the data collection process, or it may be chosen heuristically by concatenating samples according to domain-specific needs. In our experiments, we choose $r=\ell+1$ only to demonstrate that our algorithm can flexibly handle settings where the chosen rank at each time point exceeds the number of observed samples at that time point. This setting is infeasible for methods that estimate local subspaces independently at each time point: when $\ell < r$, the observation matrix at a single time point has rank at most $\ell$, making a rank-$r$ local subspace estimate infeasible without additional structure or coupling across time.
>
> > **“please clarify what is meant by a ‘non-trivial null space.’”**
>
> **Response:** A trivial null space is one that only contains the zero vector.
>
> > **“In Algorithm 1, please explicitly state how B_{i,j} is constructed.”**
>
> **Response:** Algorithm 1 refers to Equation (15) that describes how B_{i,j} is constructed.
>
> > **General changes**
>
> **Response:** Following the minor comments, we have also made the following revisions to our paper:
> - Changed the caption of Figure 1 to: $\mathbf{H}_1,\ldots,\mathbf{H}_4$ are orthonormal bases that represent subspaces (i.e., points on the Grassmannian)
> - Explicitly mentioned $r \leq n$
> - Defined the subspace $\mathbf{U}$ around Equation (2)

---

### Review · Reviewer_7QYn · 2026-06-05

**Summary Of Contributions:**

The paper studies dynamic subspace estimation from undersampled or compressive measurements. The main idea is to constrain the time-varying subspaces to lie on a single geodesic on the Grassmann manifold, and to estimate the latent signals and geodesic parameters through a Riemannian block majorize-minimize algorithm. The authors also propose a spectral initialization scheme and provide convergence guarantees in terms of monotonic objective decrease and convergence to an (\epsilon)-stationary point. Experiments are conducted on synthetic data, video reconstruction, and dynamic fMRI reconstruction.

The paper addresses a meaningful problem. Modeling time-varying low-rank structure is important, especially when only few or incomplete measurements are available at each time point. The use of a Grassmannian geodesic gives a clean geometric parameterization, and the empirical results suggest that this model can be useful in some undersampled dynamic settings.

The main weaknesses are that the novelty is not fully clear, the single-geodesic assumption is quite strong, and some claims about faithfully estimating best-fit subspaces appear stronger than what is proved or empirically verified. The experiments are encouraging but still rely heavily on planted geodesic settings or reconstruction metrics rather than direct subspace recovery on real data. I also have concerns about whether all assumptions required by the convergence proof are fully checked.

**Additional Comments:**

Overall, the paper studies an interesting problem and contains useful ideas. The geometric model is elegant, and the empirical results are encouraging. However, I do not think the current version fully supports the strongest claims about faithful subspace recovery. The main contribution also needs clearer positioning relative to existing Grassmannian geodesic and dynamic subspace tracking literature. I would recommend substantial revision before acceptance.

**Audience:**

Yes

**Audience Explanation:**

The paper is relevant to several communities within TMLR's scope, including low-rank modeling, inverse problems, subspace tracking, Riemannian optimization, and learning from incomplete or compressive measurements. The idea of using a geometric trajectory model on the Grassmann manifold for dynamic low-rank recovery is interesting, and the applications to video and dynamic fMRI reconstruction are potentially useful. Even if the current version needs revision, the problem setting and empirical observations would likely be of interest to readers working on dynamic representation learning, matrix recovery, and optimization on manifolds.

**Broader Impact Concerns:**

I do not see major ethical or societal concerns specific to this work. The paper is primarily methodological and focuses on dynamic subspace estimation and reconstruction from incomplete measurements. If the authors discuss applications such as medical imaging, they should make clear that the method is intended for reconstruction/research use and that clinical deployment would require additional validation.

**Claims And Evidence:**

No

**Claims Explanation:**

The empirical results support the claim that the proposed model can be useful for reconstruction under certain dynamic and undersampled settings. The synthetic experiments show good subspace recovery when the data are generated from the assumed geodesic model, and the video/fMRI experiments show promising reconstruction quality.

However, some claims are not yet fully supported. In particular, the statement that the method can “faithfully estimate the best-fit subspaces at each time point” seems too strong. The theory proves monotonic decrease and convergence to an (\epsilon)-stationary point, but it does not prove statistical consistency, identifiability, sample complexity, or recovery of the true/best-fit subspaces. The strongest subspace recovery evidence comes from planted synthetic experiments where the model assumption is exactly satisfied. For real video and fMRI data, the evaluation is mainly based on reconstruction PSNR/NRMSE rather than direct validation of the estimated subspaces.

The paper also assumes that the subspace trajectory follows a single global Grassmannian geodesic. This is elegant, but may be restrictive for real dynamic data, where the subspace path may be nonlinear, piecewise smooth, or affected by abrupt changes. The paper does not sufficiently analyze what happens under model mismatch.

Finally, the convergence proof relies on a general RBMM theorem, but several required assumptions appear to need more careful verification, including the validity of the global majorizers, the summability of optimality gaps, and the treatment of inexact surrogate minimization. The theory assumes positive proximal parameters, while the experiments set them to zero, so the theoretical guarantee does not exactly cover the implemented algorithm.

**Requested Changes:**

Critical changes needed for acceptance:

1.	Clarify the novelty relative to prior work. Dynamic subspace estimation, subspace tracking, Grassmannian geodesic models, and Riemannian optimization have all been studied before. The paper should explain more precisely what is new beyond applying a single-geodesic Grassmannian model to undersampled data with a block MM solver. The relation to prior Grassmannian geodesic subspace estimation/tracking methods should be expanded, and any closely related formulations should be discussed explicitly.

2.	Weaken or justify the claim of “faithfully estimating the best-fit subspaces.” The current theory does not establish subspace recovery, identifiability, or sample complexity. The authors should either weaken this claim or provide additional theory/experiments that directly validate best-fit subspace estimation beyond the planted geodesic setting.

3.	Analyze the single-geodesic assumption more carefully. The paper should test model mismatch: non-geodesic subspace paths, piecewise-geodesic paths, abrupt changes, inaccurate time parameters (t_i), and different choices of rank and regularization. This is important because the model may be too restrictive for many real dynamic datasets.

4.	Strengthen the experimental comparisons. The current baselines are mostly static low-rank methods such as AltMin/GNMR, with limited dynamic comparisons. The paper should include stronger dynamic baselines, missing-data subspace tracking methods, or piecewise/regularized dynamic low-rank models if possible. Comparisons with simpler solvers under the same geodesic model, such as Riemannian gradient descent, alternating minimization, or trust-region methods, would also help justify the RBMM algorithm.

5.	Improve ablation studies. More ablations are needed for initialization, rank (r), number of samples per time point (\ell), sampling ratio, noise level, regularization parameter (\lambda), and the range/choice of (t_i). These ablations would clarify when the method works and when it fails.

6.	Make the proof more rigorous. The monotonicity proof should explicitly show that each proposed surrogate is a valid global majorizer. The convergence proof should carefully verify all assumptions required by the cited RBMM theorem, especially the summability of optimality gaps and the inexact computation condition. The (Q)-update should distinguish gradients evaluated at (Q) and (Q^k), and the (\Theta)-update should handle edge cases such as (t_i=0) and possible denominator singularities. Since the theory assumes positive proximal parameters but the experiments use zero proximal parameters, the authors should either extend the theory or align the implementation with the assumptions. Stationarity should also be stated in terms of the appropriate Riemannian or projected gradient.

Changes that would strengthen the work:

7.	Report runtime, memory, and scalability comparisons, especially as (n), (T), (r), and the sampling ratio vary.

8.	Provide more direct subspace evaluation on real or semi-real data where approximate ground-truth subspaces can be estimated.

9.	Discuss failure cases. For example, when does the geodesic model introduce artifacts or perform worse than static/piecewise methods?

10.	Improve some notation and presentation details. Some formulas and algorithmic descriptions should be checked carefully, including the augmented operator notation, the SVD-based initialization, and the exact dimensions of variables.

---

> ### Author Response · Authors · 2026-06-12
> **Author Response (Part 1)**
>
> Thank you for taking the time to review our paper. Below, we address the concerns in detail:
>
> > **"Clarify the novelty relative to prior work… The relation to prior Grassmannian geodesic subspace estimation/tracking methods should be expanded."**
>
> **Response:**
>
> - Thank you for raising this point. Existing methods such as GROUSE also use geodesics, for example in their stochastic gradient updates, as do many manifold gradient descent algorithms. However, in these methods, the geodesic is primarily an optimization tool: each update follows a different geodesic determined by the current gradient direction. In contrast, our goal is to model the time-varying subspace itself as a geodesic curve on the Grassmannian.
> - One motivation for studying this batch geodesic learning problem was precisely that methods such as GROUSE optimize a fixed objective in a streaming or online manner, and this streaming aspect is often used to handle slowly time-varying data. We instead take a step back and ask what a natural model for time-varying subspaces should be. This leads us to view the evolving subspace as a curve on the Grassmannian. While a piecewise-geodesic approximation is a natural approach, the first step is to understand how to optimize a data-fitting objective over a single geodesic, which is the focus of our work. We have revised the manuscript to better reflect this motivation and to clarify the differences between our formulation and prior Grassmannian subspace estimation and tracking methods.
> - Please let us know if there is any particular relationship you would like us to expound on, and we will be happy to do so.
>
> > **"Weaken or justify the claim of “faithfully estimating the best-fit subspaces.”**
>
> **Response:** We agree that “best-fit subspaces” conjures a stronger claim that our method would find the best-fit geodesic curve to arbitrary data. Instead we have edited so that the contributions are clear: We developed an algorithm for a non-convex formulation of the problem of finding a best-fit geodesic. We showed plentiful evidence that in synthetic experiments in the planted setting, with data that comes from a noisy geodesic, our algorithm succeeds in recovering the planted geodesic. And we showed initial promising experiments in real applications where the geodesic model gives improved metrics as compared to other low-rank models.
>
> > **"Analyze the single-geodesic assumption more carefully."**
>
> **Response:** We added a new Section 5.3 that presents ablation studies on the effect of the timesteps and justifies our choice of timestep range, as well as the effect of the hyperparameter $\lambda$. We also added a baseline for the case $\ell > r$, as suggested by another reviewer, where it is possible to estimate local subspaces independently. These ablation studies clarify how the hyperparameters can be chosen in practice and further support our geodesic modeling assumptions.
>
> > **“Strengthen the experimental comparisons. The current baselines are mostly static low-rank methods such as AltMin/GNMR, with limited dynamic comparisons.”**
>
> **Response:**
> - Following the suggestion of Reviewer ayvE, we added another baseline in Section 5.3 based on the steps outlined [here](https://imgur.com/a/mgeeQcc). While our method still outperforms this baseline, we emphasize that this baseline is applicable only when $\ell \geq r$, where local subspaces can be estimated independently, and it is not straightforward to extend to settings with compressive maps.
> - We clarify in Section 2.2 that our choice to compare to these static methods is primarily motivated by two reasons:  (i) it is often nontrivial to extend existing methods to cases where the linear operator is not the identity, and (ii) many existing works assume that subspaces change one at a time and that the changes between subspaces are minimal—assumptions that are not needed by our algorithm. Nevertheless, we included an additional baseline, GROUSE, an online subspace estimation algorithm on the Grassmannian via rank-one updates. We still observe that our algorithm generally outperforms these baselines and leads to clearer visual reconstructions.
> - Furthermore, we clarify that for the fMRI data experiments, the "data sharing" reconstruction is
> is a *dynamic* reconstruction method that is a very natural baseline for this application.
>
> > **“Improve ablation studies. More ablations are needed”**
>
> **Response:** As stated earlier, we have included additional ablation studies in Section 5.3.
>
> > **“Make the proof more rigorous. The monotonicity proof should explicitly show that each proposed surrogate is a valid global majorizer.”**
>
> **Response:** As suggested, we revised the manuscript to clarify the monotonicity proof, particularly by showing that each proposed surrogate is a valid global majorizer and explicitly verifying the required properties.

---

> > ### Author Response · Authors · 2026-06-12
> > **Author Response (Part 2)**
> >
> > > **Changes that would strengthen the work**
> >
> > **Response:** Thank you for these suggestions. We have a short discussion on the cases in which the existing baselines (e.g., AltMin) outperforms our algorithm. This includes the cases in which the concatenated data is exactly low-rank, as in the Shepp-Logan phantom experiments in the Appendix. We have also clarified notation throughout.

---

### Author Response · Authors · 2026-06-12
**Summary of Main Changes**

We thank the reviewers for their thoughtful comments on our manuscript. We are pleased the reviewers noted that our work "addresses a meaningful problem" (Reviewer 7QYn), holds "both theoretical and practical relevance" (Reviewer ayvE), and is "clearly written and well structured" (Reviewer Bbv6). Below, we summarize the main revisions made to the manuscript, which are also marked in red:

> **Addition of Section 5.3:**

- We added a new section (Section 5.3), which provides additional ablation studies requested by several reviewers. These include analyzing the effect of the time steps $t_i$ and the regularization parameter $\lambda$. These results highlight how to select these hyperparameters in practice and further demonstrate the effectiveness of our geodesic modeling approach.

- Also in Section 5.3, we included a comparison to another baseline we call "tangent regression," as suggested by Reviewer ayvE. As the reviewer mentions, this is a natural baseline to consider when $\ell \geq r$, where one is able to estimate local subspaces. Even in this setting, we show that our modeling approach outperforms this baseline across all frames, yielding consistently higher PSNRs throughout.

> **Addition of GROUSE as a new baseline:**

- The reviewer also raised a concern about the lack of experimental baselines. To address this, we included GROUSE as an additional baseline alongside AltMin and GNMR. Our experimental results show that our method generally outperforms these baselines on video datasets, yielding clearer visual reconstructions.

> **Revision of proofs for more rigor:**

- Following the suggestions of Reviewers 7QYn and Bbv6, we revised the manuscript to improve the rigor of our proofs and the clarity of our notation. These revisions include resolving the ambiguous overloading of the function $f$, verifying that the properties of the majorizer in Theorem 2 hold for each block, and more carefully checking each condition in Theorem 1.

> **Clarification of novelty relative to prior work:**

- In Section 2.2, we added a paragraph highlighting the main difference between our algorithm and methods like GROUSE that also use geodesics. Specifically, we clarify that our approach explicitly models the time-varying subspace itself as a geodesic curve on the Grassmannian.

---

### Decision · Action_Editor_HvAi · 2026-07-06

**Recommendation:** Accept as is

**Audience:**

Yes

**Audience Explanation:**

All the reviewers agree that the work would be of interest to part of the TMLR audience.

**Claims And Evidence:**

Yes

**Claims Explanation:**

Two out of three reviewers argue that the claims are sufficiently supported, thus recommending acceptance, whereas the third reviewer expresses some concerns, mostly related to the assumption that the time-varying subspace lies on a single geodesic, thus rating the paper as a borderline reject. The limitation of a single geodesic was also highlighted by another reviewer. Nevertheless, considering other positive aspects of the paper, and the fact that all reviewers were mostly pleased with the revision and authors' feedback, the AE believes that the paper can be accepted. The limitation is acknowledged in the conclusion.